# A receptor for the complement regulator factor H increases transmission of trypanosomes to tsetse flies

Olivia J.S. Macleod [1], Jean-Mathieu Bart [2], Paula MacGregor [1], Lori Peacock[3], Nicholas J. Savill [4], Svenja Hester[5], Sophie Ravel[2], Jack D. Sunter [6], Camilla Trevor[1,7], Steven Rust[7], Tristan J. Vaughan[7], Ralph Minter [7], Shabaz Mohammed [5], Wendy Gibson[3], Martin C. Taylor [8], Matthew K. Higgins [5]✉ & Mark Carrington [1]✉

Persistent pathogens have evolved to avoid elimination by the mammalian immune system including mechanisms to evade complement. Infections with African trypanosomes can persist for years and cause human and animal disease throughout sub-Saharan Africa. It is not known how trypanosomes limit the action of the alternative complement pathway. Here we identify an African trypanosome receptor for mammalian factor H, a negative regulator of the alternative pathway. Structural studies show how the receptor binds ligand, leaving inhibitory domains of factor H free to inactivate complement C3b deposited on the trypanosome surface. Receptor expression is highest in developmental stages transmitted to the tsetse fly vector and those exposed to blood meals in the tsetse gut. Receptor gene deletion reduced tsetse infection, identifying this receptor as a virulence factor for transmission. This demonstrates how a pathogen evolved a molecular mechanism to increase transmission to an insect vector by exploitation of a mammalian complement regulator.

[1] Department of Biochemistry, University of Cambridge, Tennis Court Road, Cambridge CB2 1QW, UK. [2] Intertryp, IRD, Cirad, University of Montpellier, Montpellier, France. [3] School of Biological Sciences, University of Bristol, Bristol BS8 1UG, UK. [4] Institute for Immunology and Infection Research, School of Biological Sciences, University of Edinburgh, King's Buildings, West Mains Road, Edinburgh EH9 3JT, UK. [5] Department of Biochemistry, University of Oxford, South Parks Road, Oxford OX1 3QU, UK. [6] Department of Biological and Medical Sciences, Oxford Brookes University, Gipsy Lane, Oxford OX3 0BP, UK. [7] Department of Antibody Discovery and Protein Engineering, AstraZeneca R&D, Granta Park, Cambridge CB21 6GH, UK. [8] Faculty of Infectious and Tropical diseases, London School of Hygiene and Tropical Medicine, London WC1E 7HT, UK. ✉email: matthew.higgins@bioch.ox.ac.uk; mc115@cam.ac.uk

The mammalian complement system comprises abundant plasma proteins that initiate and propagate processes on pathogen surfaces resulting in their elimination. The complement cascade is highly conserved across mammals and has three arms: (i) the classical pathway triggered by specific binding of antibodies or complement components to foreign antigens, (ii) the lectin pathway initiated by binding of lectins to foreign oligosaccharides, and (iii) the alterative pathway initiated by spontaneous and constitutive deposition of C3b onto a cell surface[1–3]. All pathways trigger a response that can result in opsonization for phagocytosis and/or the formation of a membrane attack complex (MAC) that kills pathogens by compromising the integrity of the plasma membrane.

Complement activation is limited on self-cells by a number of negative regulators. For the alternative pathway, self-cells are protected by complement factor H (FH), which binds to components of the negatively charged glycocalyx[4]. FH blocks the complement cascade by accelerating the decay of C3 convertase (C3bBb), thus reducing C3b production, and by acting as a co-factor for factor I, which cleaves and inactivates C3b[4]. FH is composed of 20 tandem complement control protein (CCP) domains with 6, 7 and 20 binding self-cell-negative surface markers, whereas 1–4 and 19 bind C3b[5–8].

FH recruitment by diverse pathogens has been shown to be an effective strategy to inactivate complement in the host, but the molecular basis has only been characterized for three bacteria: *Streptococcus pneumoniae*, *Borrelia burgdorferi* and *Neisseria meningitidis*[9–12]. These recruit FH by mimicry of mammalian host interactions with FH or by increasing activity of FH. The causal agent of malaria, *Plasmodium falciparum*, recruits FH at three points in its developmental cycle, but molecular mechanisms have not yet been determined[13–15].

In this work, we describe the identification and characterization of a *Trypanosoma brucei* FH receptor (FHR). *T. brucei* is an extracellular protozoan pathogen that causes human and animal trypanosomiasis and is transmitted by tsetse flies[16,17]. *T. brucei* has a complex life cycle with a series of developmental forms, each having evolved a specialized cell surface to counteract host defences in the relevant niche[18,19]. Although reported cases of the human disease have diminished in the last decade, the animal disease acts both as a reservoir of human infective trypanosomes[20] and continues to reduce livestock production, representing one of the largest constraints on livestock productivity by pastoralists[21].

The mechanisms by which *T. brucei* counteracts the mammalian adaptive immune response are well-characterized: antigenic variation at the population level and rapid clearance of surface-bound immunoglobulin at the individual cell level[22–24]. In addition, the pathways that inactivate trypanolytic factors, a specialized form of innate immunity unique to humans and a few other primates, have been characterized[25,26]. However, although *T. brucei* activates the alternative complement pathway, it is not known how progression to the MAC is prevented[27–29]. Here we identify a trypanosome receptor that binds mammalian FH and understand the molecular basis for the interaction, revealing how a parasite exploits a mammalian protein to increase transmission to an insect vector, a strategy that is likely to have evolved independently many times in pathogens.

## Results

**Identification of a *T. brucei* FHR**. Only two African trypanosome receptors for host macromolecules have been functionally characterized: the transferrin receptor and the haptoglobin haemoglobin receptor[30,31]. This work started with an assumption that the interactions between trypanosomes and their hosts are likely to be more extensive. A screen of the *T. brucei* genome was performed to identify putative receptors based on one or both of two criteria: first, a prediction that the structure contained a three-helical bundle core, common in other characterized trypanosome surface proteins[32] and, second, that a cell surface localization was likely. The outcome was a list of 13 genes/gene families (Supplementary Table 1).

One of these, Tb927.5.4020, encodes a polypeptide of 227 residues including predicted N-terminal signal and C-terminal glycosylphosphatidylinositol (GPI)-anchor addition sequences. The predicted mature polypeptide was expressed as a glutathione-S-transferase (GST) fusion protein, immobilized on glutathione beads, and used to precipitate any potential ligand from bovine serum. The most abundant polypeptide was 145 kDa and was identified as complement FH by mass spectrometry (MS) (Fig. 1a and Supplementary Fig. 1a). The interaction was confirmed using surface plasmon resonance (SPR). The putative receptor was produced with a single biotinylation site near the C terminus and was immobilized on a streptavidin chip. It bound bovine FH, either purified from serum or as a recombinant protein, with a $K_D$ value of 153 nM and 114 nM, respectively (Fig. 1b and Supplementary Fig. 1b-f). FH is a highly abundant protein, with concentrations ranging from 1.6 to 4.9 µM in normal human adult serum[33] and is similarly abundant in bovine serum[34]. This concentration would enable near-to-complete saturation of the receptor at the measured affinity. Together, these experiments identify Tb927.5.4020 as a *T. brucei* FHR.

**FHR membrane-distal region interacts with FH domain 5**. Next, the interaction interface between FHR and FH was characterized. The binding site on FHR was investigated by identifying conserved residues in the closest homologues to FHR in the related species *Trypanosoma congolense* and *Trypanosoma suis*. An alignment of these sequences highlighted a single region close to the mature N terminus with most residues identical in all three (Supplementary Fig. 1g). The conserved residues were tested by mutagenesis of FHR followed by pulldowns from serum to assess the effect of mutations on FH binding (Supplementary Fig. 1h). FH was detected in pulldowns with FHR wild type (WT) and mutant 1 (K46A, R47A), but was not detected with mutant 2 (E31A, Q34A) and mutant 3 (E31A, Q34A, K46A, R47A) (Supplementary Fig. 1i). Circular dichroism (CD) spectroscopy was used to confirm that the changes made in mutants 2 and 3 had not substantially altered the structure at the temperatures at which experiments were performed (Supplementary Fig. 2a-d). Therefore, mutants 2 and 3 disrupted FH binding and were used for further analysis.

A chemical cross-linking analysis was performed to identify which part of FH bound FHR. FHR WT or non-binding mutant 3 were both biotinylated at the C terminus and mixed with FH and a concentration range of either H12 or D12 disuccinimidyl suberate (DSS) (Fig. 1c, d and Supplementary Fig. 2e, f). Specific cross-linking was demonstrated by the presence of high-molecular-weight biotinylated polypeptides in the reaction containing the WT but not with mutant 3 (Fig. 1c, d and Supplementary Fig. 2e, f). Cross-linked residues were identified by MS of tryptic peptides. The most abundant cross-link by far was between residue 283 in domain 5 of FH and residue 24 near the membrane-distal N terminus of FHR (Supplementary Fig. 2g). Other less abundant cross-links from FHR to FH domains 8 and 20 were also detected (Supplementary Fig. 2g).

These findings were used to guide the selection of FH domains, expressed as single or tri-domains, for the investigation of FHR binding specificity and kinetics (Fig. 2 and Supplementary Fig. 3a-c). The highest affinity interaction, with a similar $K_D$ to

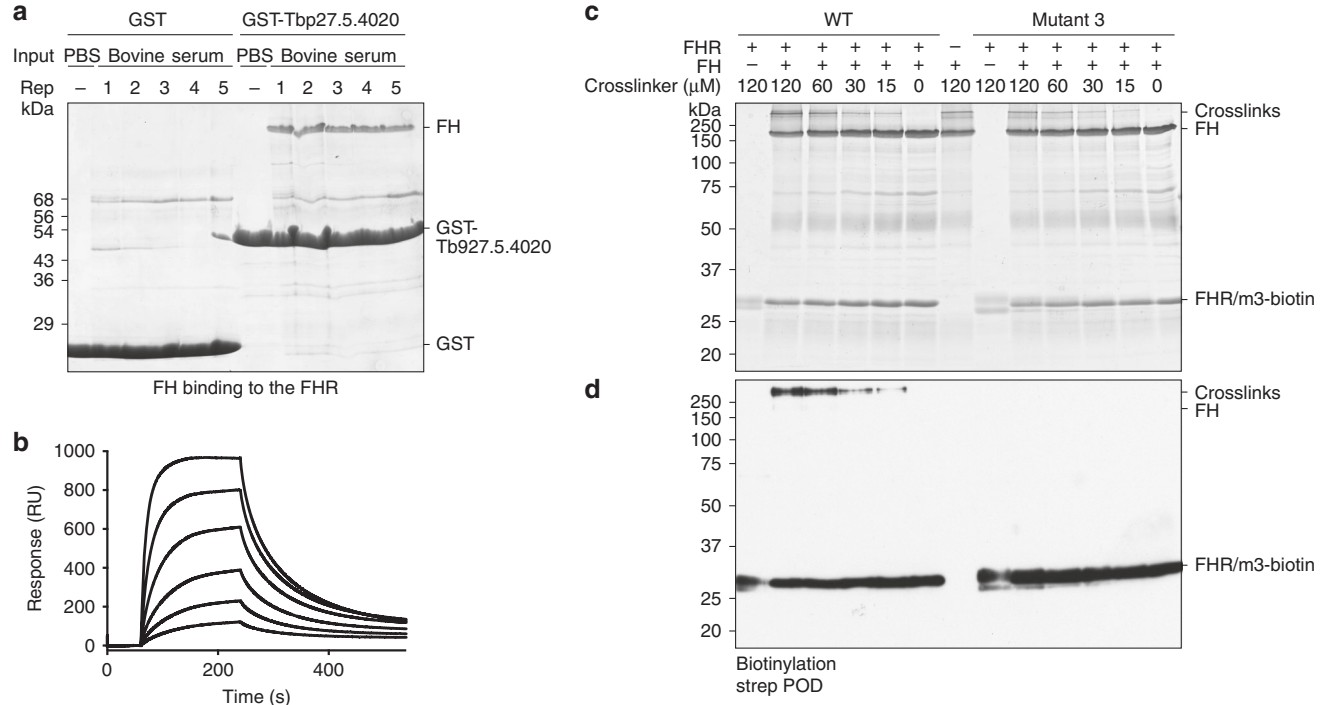

**Fig. 1 Identification of the *T. brucei* FHR and its interaction interface with FH. a** SDS-PAGE analysis of pulldowns from bovine serum were performed with GST or GST-Tb927.5.4020 immobilized on beads in five replicates or with a phosphate buffered saline (PBS) control. **b** SPR-binding data for C-terminally biotinylated FHR (450 RU) bound to a streptavidin chip and twofold dilutions of FH purified from bovine serum (1 μM highest concentration). Data shown are representative of three repeats. **c** SDS-PAGE analysis of cross-linking experiment using D12 disuccimidyl suberate (DSS) to probe the interaction between FHR and FH. FHR and mutant 3 were biotinylated at the C terminus and incubated with FH prior to the addition of the cross-linker in a dilution series as shown. **d** Western blotting of samples in **c**, probed with streptavidin peroxidase. Source data are provided as a Source Data file.

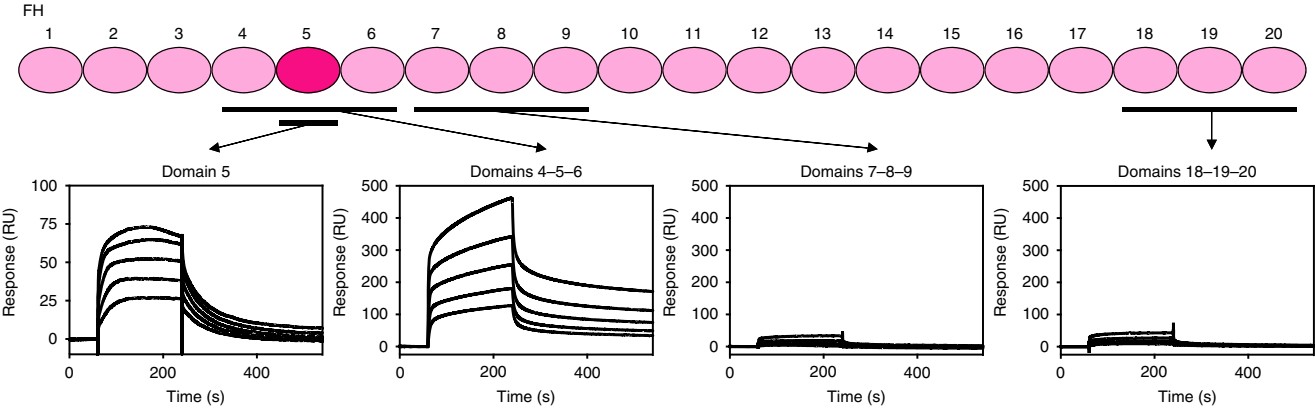

**Fig. 2 FHR binds FH domain 5.** Schematic of the 20 domains of bovine FH showing the domains that were assessed for binding to FHR with the SPR-binding data for each shown below. C-terminally biotinylated FHR (400 RU) was bound to a streptavidin chip and twofold dilutions of FH domains 4-5-6 (1.5 μM highest concentration), 7-8-9 (30 μM highest concentration), 18-19-20 (40 μM highest concentration), and domain 5 alone (5 μM highest concentration) were analysed. Source data are provided as a Source Data file.

full-length FH, was between FHR and FH domains 4-5-6 with a $K_D$ of 167 nM and the $K_D$ for binding to domain 5 alone was 562 nM (Supplementary Fig. 3b, c). Binding of FHR to domains 7-8-9 and 18-19-20 were too weak to allow calculation of $K_D$ values from the data obtained. Therefore, the N-terminal region of FHR interacts primarily with FH domain 5.

**FHR holds FH in a position conducive to C3b inactivation.** The structure of the FHR bound to FH domain 5 was determined by X-ray crystallography (Fig. 3, Supplementary Fig. 3d-h and Tables 2 and 3). The ordered part of the FHR structure observed in the crystal starts at the mature N terminus (residue 25) and

ends 14 residues before the predicted mature C terminus and GPI anchor (residue 237). FHR is an elongated three-helical bundle of ~90 Å similar to other trypanosome receptors[32,35], but has evolved a binding pocket for FH domain 5 through spreading apart of the membrane-distal ends of the long second and third helices (Fig. 3 and Supplementary Table 3). Domain 5 has a CCP fold as expected[36,37], with its N terminus pointing away from the plasma membrane (Fig. 3a). Two copies of FHR and two of FH domain 5 were present in the asymmetric unit of the crystal (Supplementary Fig. 3f, g) with each receptor contacting both copies of domain 5. Mutant 2 FHR (Supplementary Fig. 1h) resolved which interaction was representative of the physiological

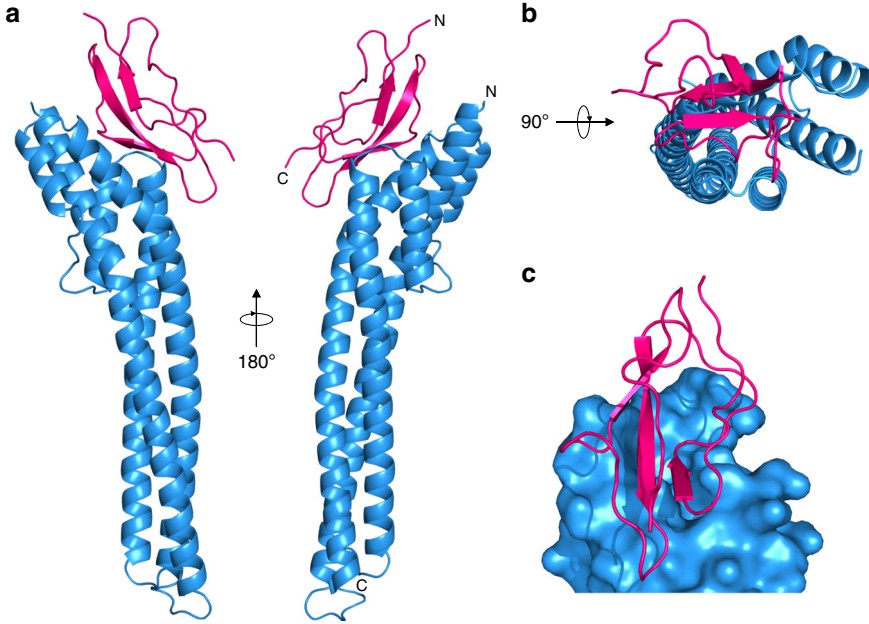

**Fig. 3 The structure of FHR bound to FH domain 5. a** Two views of FHR (blue) bound to FH domain 5 (pink). The N and C termini of FHR and domain 5 are indicated. The C terminus of FHR is plasma membrane proximal. **b** View of FHR (blue) bound to FH domain 5 (pink) viewed from the membrane-distal end. **c** Surface representation of the FH-binding pocket in FHR (blue) with the FH domain 5 (pink).

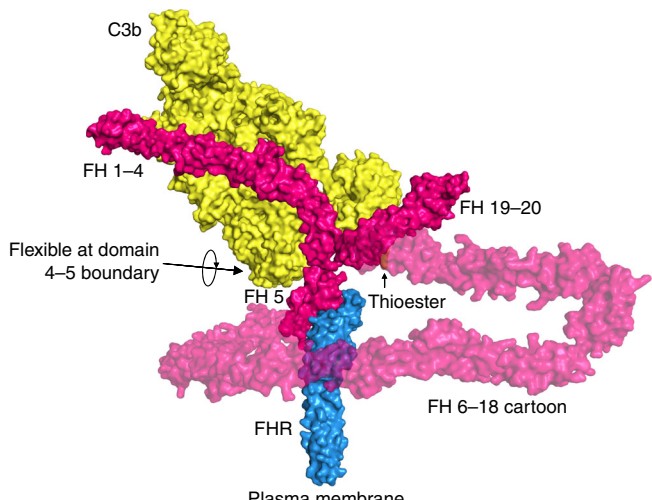

**Fig. 4 FHR holds FH in a position in which it can access surface-bound C3b.** A cartoon model for how the FHR (blue) could hold FH (pink) distal to the plasma membrane, to interact with C3b (yellow). The model combines the FHR:FH domain 5 structure with previously determined structures PDB 5O35 and 2QFG.

complex and which was due to crystal packing. That is, mutant 2 contained mutations E31A and Q34A within one of the possible interfaces; these alterations disrupted the receptor-ligand interaction by pulldown (Supplementary Fig. 1i) and also by SPR analysis (Supplementary Fig. 3d, e), thus confirming that the interaction is mediated by the binding site containing these residues shown in Fig. 3.

The single predicted N-linked glycosylation site in FHR is located at the extreme membrane-distal tip of the molecule (Supplementary Fig. 3i) and in this position the oligosaccharide could potentially reduce antibody accessibility of FHR molecules embedded in the trypanosome surface coat without interfering with FH binding.

To understand how the interaction might affect complement function, we produced a model for the FHR-FH-C3b complex (Fig. 4). This model used the known structures of C3b in complex with FH domains 1–4 linked to 19-20 (PDB 5O35)[38] and FH domains 1–5, which show the domain 4-5 boundary to be highly flexible (PDB 2QFG)[37] (Supplementary Fig. 3j). In the model of the FHR-FH-C3b complex, domains 1–4 of FH remain free in the receptor-bound form to access and bind C3b deposited on the periphery of surface coat and so inhibit the complement cascade.

FHR was identified and characterized using bovine serum and FH. As African trypanosomes can infect a wide range of mammals, we also tested whether FH from different species could bind FHR. First, the degree of conservation of the 18 interactions between FHR and bovine FH domain 5 (Supplementary Table 3) was assessed, and this varied from 18 potentially conserved interactions in sheep, 17 in goat and 9–15 in pig, horse, rabbit, rat, mouse and human (Supplementary Table 4). Next, interactions were tested directly using the pulldown approach that identified bovine FH (Supplementary Fig. 4a, b). FHR clearly pulled down FH from sheep, goat and pig sera, but FH from horse, rabbit, rodent and human was not readily observed, suggesting weaker interaction with FHR. The likelihood of pulldown broadly followed the number of conserved interacting residues.

As pulldown outcomes are very sensitive to off rates, FH from two species with low numbers of conserved interactions, human and mouse, were made as domains 4-5-6 recombinant proteins and tested for interaction with FHR using SPR (Supplementary Fig. 4c-e). Both mouse and human proteins bound to WT FHR but not mutant 2 FHR, albeit with weaker affinities of $K_D = 12$ μM for mouse and $K_D = 8.7$ μM for human, compared with that of bovine, $K_D = 0.167$ μM. These affinities are in the same range as the concentration of FH in serum, suggesting that a fraction of the receptors will be FH bound in vivo. A further SPR experiment was performed using immobilized FHR and mouse plasma from WT and FH-null mice ($Cfh^{-/-}$) (Supplementary Fig. 4f). There was a clear binding response of 200 RU from the WT serum but not the FH-null plasma. Together, these observations indicate

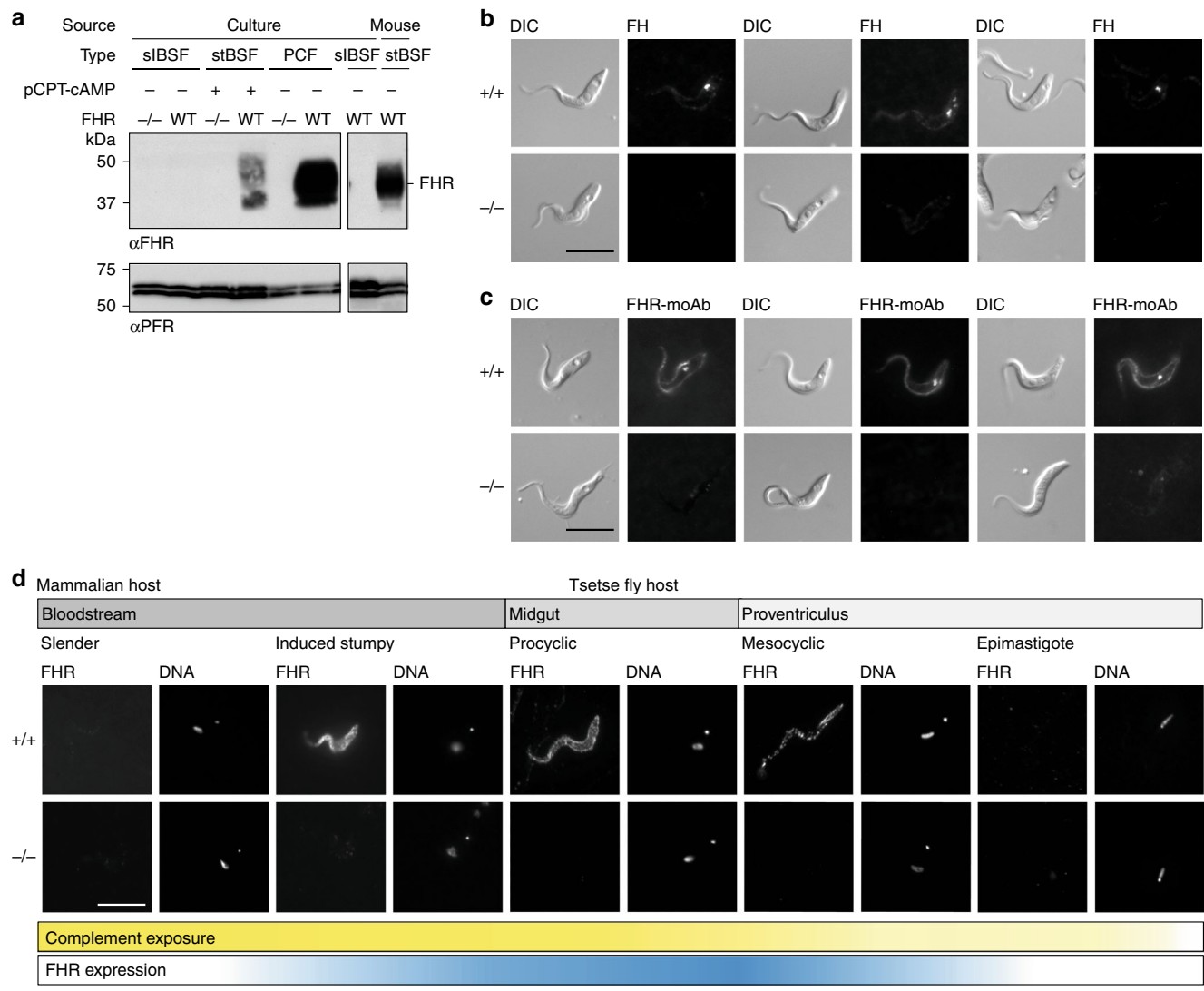

**Fig. 5 FHR is upregulated on the cell surface of forms that are adapted to life in blood ingested by the tsetse fly. a** Western blot analysis probed with FHR antiserum (αFHR) of whole-cell lysates from FHR$^{+/+}$ and FHR$^{-/-}$ slender BSFs, induced stumpy BSFs, and PCFs in culture and from FHR$^{+/+}$ stumpy BSFs from a mouse infection. Anti-PFR (αPFR) was used as a loading control. **b, c** Live-cell-binding assay of 100 nM fluorescent **b** bovine FH or **c** FHR-moAb to PCFs in culture. After 2 h, cells were fixed in culture and visualized. Scale bar, 10 μm. **d** Immunofluorescence analysis of localization of FHR during the majority of the *T. brucei* life cycle. Cells were collected, fixed and then probed with FHR antiserum. Scale bar, 10 μm. DNA, Hoechst staining of the nuclear and kinetoplastid DNA. Cells positioned with the posterior towards the top right corner. Bars beneath the microscope images represent where complement is present during the life cycle (yellow gradient) and FHR expression (blue gradient). Source data are provided as a Source Data file.

that FHR interacts specifically with mouse FH. We therefore continued to characterize the cellular behaviour of FHR and assess its function in tsetse flies and in the standard mouse model of infection.

**FHR is expressed in tsetse-transmissible and gut forms.** In the mammalian host, *T. brucei* exists as proliferating slender and quiescent stumpy bloodstream forms (BSFs). Slender BSFs differentiate to stumpy BSFs in response to a density-dependent quorum-sensing mechanism and both stages are continually exposed to complement[39,40]. Only the stumpy BSFs readily infect tsetse flies[39,40] where, in the midgut, they differentiate first into procyclic forms (PCFs) and subsequently mesocyclic forms as they migrate towards the proventriculus[41]. The midgut and proventriculus are exposed to fresh complement in the immediate aftermath of a blood meal. Mesocyclic forms in the proventriculus differentiate to epimastigote forms, which migrate to the salivary

glands away from direct contact with blood and differentiate further to forms able to re-infect the mammalian host.

Whole-cell lysates from cultured slender and induced stumpy BSFs and PCFs were used to determine FHR expression levels by semi-quantitative western blotting. First, the specificity of FHR antiserum was shown using a set of lysates from cells with both alleles of FHR deleted (FHR$^{-/-}$) as negative controls (Fig. 5a and Supplementary Fig. 5a, b). FHR expression was detected in induced stumpy BSFs[42,43] and increased further in PCFs with an estimated ~200,000 copies/cell (Fig. 5a and Supplementary Fig. 5c, d). FHR was below the level of detection in slender BSFs (< ~200 copies/cell) (Fig. 5a and Supplementary Fig. 5c, d). The increase in expression on differentiation from slender to induced stumpy BSFs was confirmed using cell lysate of stumpy BSFs isolated from a mouse infection (Fig. 5a). Thus, expression of FHR increases during differentiation from slender to stumpy BSFs and further in PCFs where it is an abundant protein.

**FHR binds both FH and a monoclonal antibody in live cells**. To investigate FHR function and accessibility to ligand on live cells, fluorescently labelled FH or FHR monoclonal antibody (moAb) were each added to a culture of PCFs for 2 h and any binding visualized by microscopy (Fig. 5b, c). Ligands were added directly to growing cells, so labelled FH added would be diluted by FH in the serum in the culture medium. Nevertheless, in FHR$^{+/+}$ cells, signal for fluorescently labelled FH was observed both on the cell surface and within cells indicating that the FHR binds FH on the cell surface and may mediate some endocytosis (Fig. 5b). FHR$^{+/+}$ cells also bound the FHR-moAb over the entire cell surface, with a fraction endocytosed (Fig. 5c). No binding of either ligand was observed to FHR$^{-/-}$ cells (Fig. 5b, c). These observations demonstrate that FHR on the cell surface is accessible to large ligands and supports the structural model of FHR-mediated protection against complement killing.

**FHR is on the cell surface of forms exposed to blood**. The subcellular localization of FHR was investigated by immuno-fluorescence. First, FHR was distributed over the entire cell surface of induced stumpy BSFs (Fig. 5d and Supplementary Fig. 6). Second, as developmental forms in the tsetse fly are not readily accessible in culture, trypanosomes from infected flies were used to assess expression in the various developmental forms in the gut. Flies were infected with PCFs of the parental FHR$^{+/+}$ and FHR$^{-/-}$ clones in the absence of complement. All cell lines established infections without significant difference between FHR$^{+/+}$ and FHR$^{-/-}$ clones (Supplementary Fig. 5e). At days 3 and 10 post infection, FHR was distributed over the cell surface of FHR$^{+/+}$ PCFs (Fig. 5d and Supplementary Fig. 7). By day 10 post infection, cells had migrated to the proventriculus and differentiated further[44]. Mesocyclic FHR$^{+/+}$ cells expressed FHR over the cell surface (Fig. 5d and Supplementary Fig. 8). However, FHR was not detected in epimastigote forms from the proventriculus (Fig. 5d and Supplementary Fig. 8), which is the stage that migrates to the salivary glands[44]. No signal was present throughout for the FHR$^{-/-}$ control (Fig. 5d and Supplementary Figs. 6–8). Therefore, expression of high levels of FHR is specific to stumpy BSFs and developmental forms in the tsetse fly that are exposed to mammalian blood. The cell surface localization in these forms is again consistent with FHR facilitating inactivation of complement at the trypanosome cell surface.

**FHR$^{-/-}$ *T. brucei* are attenuated in mice**. In all the experiments described above, complement had been inactivated in culture media and was absent in feeds of tsetse flies. The FHR$^{-/-}$ cell lines were made by first deleting both alleles of the FHR gene in cultured PCFs in the absence of complement (Supplementary Fig. 5a, b). Two independent FHR$^{-/-}$ PCF clones and a parental PCF FHR$^{+/+}$ were used to infect tsetse flies, again in the absence of active complement. Infection with all three trypanosome lines progressed to the salivary glands and infected glands were used to inoculate BALB/c mice. BSFs were recovered from all infected mice, indicating growth of FHR$^{-/-}$ cells in the presence of mouse alternative complement pathway. To test whether any alterations in growth rate had been caused by the FHR gene deletion, one BSF FHR$^{+/+}$ and one BSF FHR$^{-/-}$ clone were adapted to growth in culture. Both PCF and BSF FHR$^{-/-}$ cells had no growth rate phenotype (Supplementary Fig. 5f, g). These were the PCF and BSF cell lines used in the experiments above that characterized FHR expression.

Next, any phenotype caused by deletion of the FHR in the presence of mouse complement was tested using the standard mouse model of infection. Groups of five immunosuppressed and five untreated BALB/c mice were infected with BSF FHR$^{+/+}$ or

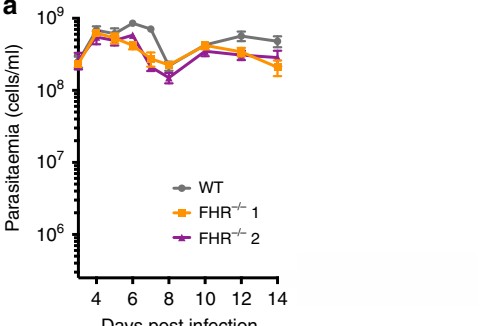

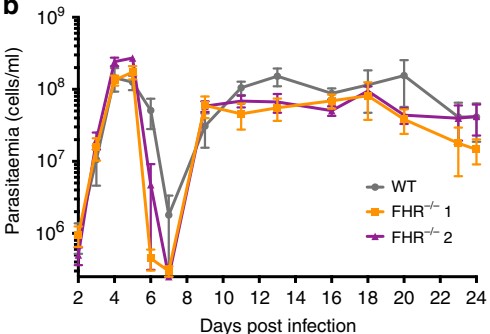

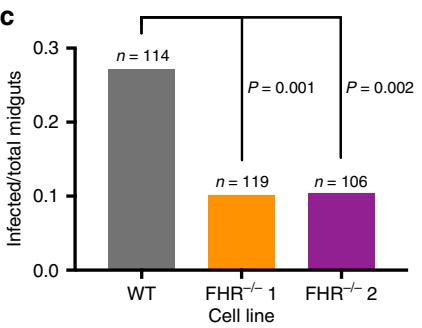

**Fig. 6 The transmission of *T. brucei* FHR$^{-/-}$ cell lines to tsetse flies from mice was significantly reduced compared with FHR$^{+/+}$. a**, **b** Mean mouse blood parasitaemia of parental FHR$^{+/+}$ and two independent FHR$^{-/-}$ clones, with error bars representing the SEM. **a** Five immunosuppressed BALB/c mice were infected with each cell line and monitored from days 3–14 post infection. **b** Five immunocompetent BALB/c mice were infected with each cell line and monitored from days 2–24 post infection. **c** Tsetse flies were fed on infected mice and transmission of the infection was evaluated by the dissection of >100 midguts per cell line on days 6–16 post infection. The total number is derived from seven replicates per cell line. The number of infected midguts over the total dissected midguts is plotted, whereby 31/114 were infected for FHR$^{+/+}$, 12/119 for FHR$^{-/-}$ clone 1, and 11/106 for FHR$^{-/-}$ clone 2. The observed difference between the FHR$^{+/+}$ and each of the FHR$^{-/-}$ clones was significant ($P < 0.002$ for each cell line by $\chi^2$-test). Source data are provided as a Source Data file.

one of the two independent FHR$^{-/-}$ clones, and blood parasitaemia was measured over a time course. FHR$^{-/-}$ clones were able to establish and maintain infections in all mice; however, a mild attenuation in the parasitaemia was observed for both FHR$^{-/-}$ clones (Fig. 6a, b and Supplementary Fig. 9a, b). In both sets of infections, the first peak of parasitaemia for FHR$^{-/-}$ clones declined before the parental FHR$^{+/+}$ and the total parasitaemia for both FHR$^{-/-}$ clones was lower at nearly all time points in the chronic phase of infection after day 10 (Fig. 6a, b and Supplementary Fig. 9a, b).

The parasitaemia in animals describes the total number of both slender and stumpy BSFs, and is influenced by the proliferation of slender BSFs, the lifespan of each life stage and the clearance of parasites by the host immune system. The upregulation of expression of FHR in stumpy BSFs (Fig. 5a, d and Supplementary Fig. 6) suggests that FHR deletion would likely cause attenuation and decreased viability of stumpy forms. A functional role for FHR in stumpy BSFs could explain both the more rapid decline in the first peak of parasitaemia and the reduced levels after the infection has established. This early decline in the first peak of parasitaemia was reminiscent of infections previously observed during mouse infections with akinetoplastic *T. brucei* caused by a reduction in stumpy BSF lifespan[45]. Using a similar mathematical model for in-host dynamics during trypanosome infections[45,46], a prediction from an analysis of the measured parasitaemia indicated that the lifespan of FHR$^{-/-}$ stumpy forms was reduced from 27 to 7 h, but this was not tested directly, as FHR is not necessary for infectivity (Supplementary Figs. 9c, d and 10, and Supplementary Data 1).

**FHR$^{-/-}$ *T. brucei* have reduced transmission to tsetse flies**. To test for the effect of FHR on transmission in the presence of complement, tsetse flies were given a single feed on mice infected with either FHR$^{+/+}$ or FHR$^{-/-}$ trypanosomes in seven replicates, in each case using mice with similar levels of parasitaemia. The tsetse flies were then fed at 2-day intervals with whole sheep blood through a feeding membrane. The flies were dissected between 6 and 16 days after the feed on infected mice and the midguts scored for the presence of trypanosomes. Both FHR$^{-/-}$ clones had a significantly reduced ability to infect tsetse flies with 2.7-fold (clone 1, $p < 0.001$ by $\chi^2$-test) and 2.6-fold (clone 2, $p < 0.002$ by $\chi^2$-test) decreases compared with the FHR$^{+/+}$ control (Fig. 6c). These observations show that FHR acts as a virulence factor for transmission to tsetse flies.

## Discussion

How trypanosomes counteract the alternative complement pathway has been little studied. Previously, it has been shown that C3 convertase (C3bBb) can assemble on the surface of BSFs but the complement cascade did not progress further and there was no recruitment of complement proteins C5b-9 to form a MAC[28]. Here we have identified and characterized FHR, a receptor that binds FH in a manner that would enable FH to negatively regulate C3b and C3bBb preventing MAC formation. FHR was not detected in proliferating slender BSFs but was upregulated in quiescent stumpy BSFs, the developmental form adapted for transmission to tsetse flies. It was further upregulated in the developmental forms exposed to blood meals in the tsetse gut, then becoming undetectable as the trypanosomes departed the midgut for the salivary glands. This FHR expression pattern is consistent with a role in transmission to and maintenance within the tsetse fly, and indeed FHR$^{-/-}$ trypanosomes had reduced transmission.

There are three steps necessary for successful transmission: first, sufficient numbers of stumpy BSFs in a blood meal; second, successful differentiation to PCFs; and third, survival and proliferation of PCFs in the face of subsequent blood meals. Modelling of the measured infection kinetics suggested that FHR$^{-/-}$ stumpy BSFs do not survive as long as WT in mice. Reduced survival may arise from differences in cell surface architecture between slender and stumpy BSFs, resulting in a greater deposition of C3b on stumpy BSFs and thus increased susceptibility to the alternative pathway that is mitigated by FHR expression. As yet, no comprehensive comparison of slender and stumpy BSF surfaces is available. Any reduced survival is unlikely to result

from an increased persistence of individual surface C3b molecules as previous work demonstrated that there is more rapid internalization and clearance of surface-bound antibodies in stumpy BSFs compared with slender BSFs[22]. Further, stumpy BSFs are more resistant than slender BSFs to antibody-dependent lysis by the classical complement pathway[22,47]. A reduced survival of FHR$^{-/-}$ stumpy BSFs may contribute to reduced transmission.

The transition of BSFs to PCFs involves a total remodelling of the cell surface and occurs in a blood meal[18,48], and FHR may contribute to protection of these differentiating cells. FHR expression increases further in trypanosomes resident in the tsetse gut. PCFs can be lysed by the alternative complement pathway in vitro and knockdown of tsetse serpins, which inactivates complement in a blood meal, reduced, but did not abolish, the ability of cultured PCFs to directly establish infections in vivo[27,29]. Expression of FHR in mesocyclic forms suggests that continued protection is favoured during and in the immediate aftermath of blood meals. The evidence here supports a model in which FHR contributes toward protection against complement but also acts in concert with tsetse fly serpins.

How are slender BSFs protected against the alternative pathway of complement given the very low expression or the absence of FHR? It is probable there are additional mechanisms of inactivation and it is likely that the trypanosome expresses a set of receptors/binding proteins for complement components, as found in various microorganisms. Further, any mechanism evolved to negate the alternative pathway is unlikely to depend on a single molecule, as this would provide a ready route for a host to evolve a countermeasure.

The structure of FHR is a further example of the three-helical bundle family of trypanosome cell surface proteins[32,35]. This structural fold clearly provides flexibility for the evolution of a range of different binding specificities. In FHR, the ends of two helices have splayed apart to form a binding pocket for domain 5 of FH. The fold is permissive for elongated receptors to access ligands within the densely packed trypanosome cell surface coat. The location of the FH-binding site is informative about the mode of action; by binding FH domain 5 at a site distal to the plasma membrane, FH domains 1–4 are located in the likely location of deposited C3b at the periphery on the cell surface coat.

FHR binds with variable affinity to FH from different mammals. *T. brucei* has a large host range and predominantly infects livestock and wild game, whereas human infective isolates are zoonotic[20]. The highest affinity of FHR was for FH from domestic animals (cow, sheep, goat, pig) most closely related to the game animals in sub-Saharan Africa that were the reservoir of *T. brucei* prior to the development of agriculture. Mouse and human FH bind with a lower affinity. Across all mammals tested, FHR-FH interactions had $K_D$ values in the high nanomolar to low micromolar range. As the concentration of FH in serum is in the low micromolar range, FHR is expected to be largely occupied, especially if there is additional avidity due to the formation of a tripartite complex of surface-bound C3b-FH-FHR.

There is more than one way to catch FH and binding proteins from a range of pathogens present a remarkable example of convergent evolution. The trypanosome FHR structure and the FH domain bound are different to each of the three structurally characterized FH-binding proteins from bacteria. These interact with FH domains 6–7 in *N. meningitidis*[12], domain 9 in *S. pneumoniae*[10] or domains 19–20 in *B. burgdorferi*[11]. The binding of these FHRs to domains 6–7 and 19–20 mimic self-surface interactions and to domain 9 produces a high-affinity interaction that enhances activity of FH for C3b in one strain[9]. Among other eukaryotic pathogens, *P. falciparum* recruits FH in blood stage schizonts and merozoites, and in emerging gametes following transmission to the mosquito midgut. In merozoites, Pf92 binds

FH through domains 5 and 20, and in gametes PfGAP40 binds FH through domains 5–7. The latter facilitates transmission to the mosquito and protection from complement in the midgut[13–15]. The structural mechanisms for the binding of FH have not been reported and the *Plasmodium* proteins are also unrelated to trypanosome FHR by sequence comparison.

Our findings represent a clear demonstration of how a pathogen receptor for a mammalian complement regulator enables it to act as a virulence factor for transmission to an insect vector. They show how trypanosomes have evolved to exploit mammalian FH and lead to a structural mechanism and mode of action of this receptor, providing a deeper understanding of pathogen anti-complement strategies.

## Methods

**Screen of genome for putative surface receptors**. A screen was performed on all *T. brucei*-annotated open reading frames (ORFs) from the TREU 927 reference genome[49] using the big-PI Predictor[50]. Each gene from this initial screen was assessed manually for the presence of an N-terminal signal sequence, the absence of a predicted transmembrane domain in the middle of the protein and a C-terminal GPI-anchor signal peptide with similarity in sequence to a pattern established by known GPI-anchored proteins (in *T. brucei*, *Ttypanosoma cruzi*, and *Leishmania major*). In addition, structural homology predictions were performed using FUGUE[51] and Phyre2[52] servers.

**GST fusion trypanosome proteins**. For the first set of pulldown experiments, DNA encoding amino acids 25–251 of the FHR was amplified from a pOPINF vector (see below) already containing this sequence. Oligonucleotides 5′-GGA TCCGGTTCTGGTAACGATAATCTTGAAGCCGAA-3′ and 5′-GTTAATCAC AAACTGGTCTAGAAAGCTT-3′ were used and the PCR product cloned in pGEM-T, sequenced, digested with BamHI and HindIII, and cloned into the corresponding sites of pGEX-KG, encoding a fusion protein with a 5-residue linker (GSGSG) between GST and FHR. In the second round of pulldown experiments with FHR and mutants, the amplicon was produced from *T. brucei* Lister 427 genomic DNA encoding residues 25–249 of the FHR using primers 5′-CATAT GGAAAATCTGTACTTCCAAGGCAACGATAATCTTGAAGCCGAATTG-3′ and 5′-GGATCCTTATTCATCTTCCTTATCTTCATCGG-3′. Product was cloned as above, this time into a modified pGEX-KG vector, whereby the resulting fusion protein had a longer linker sequence (GSSSGGG) and a Tobacco Etch Virus (TEV) protease cut site between the GST tag and FHR. Mutants 1, 2 and 3 were generated by PCR using the WT FHR vector as template and primers containing the desired mutations. To generate mutant 1: 5′- TGTGAGGTGGCGAAACAACTGGCTGC ACTTCCGCTGCTGATGGAAGAG-3′ and 5′-CTCTTCCATCAGCAGCGGAAG TGCAGCCAGTTGTTTCGCCACCTCACA-3′, and mutant 2: 5′-AACGATAATC TTGAAGCCGCATTGGAAGCGACGAAAGCACTATGTGAG-3′ and 5′-CTCAC ATAGTGCTTTCGTCGCTTCCAATGCGGCTTCAAGATTATCGTT-3′ primers were used. Mutant 3 was generated by performing PCR on a confirmed mutant 2 with mutant 1 primers.

GST fusion proteins were expressed in *Escherichia coli* BL21, incubated at 37 °C, induced for 3 h with 0.02 % (w/v) isopropyl β-D-1-thiogalactopyranoside and purified with glutathione-sepharose affinity chromatography. Purified proteins were buffer exchanged into phosphate-buffered saline (PBS).

**Pulldowns using GST fusion proteins**. Glutathione-sepharose 4B (GE Healthcare, 140 µl slurry/pulldown) was incubated with GST-tagged protein (1 mg/pulldown) for 30 min, washed with PBS and incubated with adult bovine, fetal bovine, sheep, goat, pig, horse, rabbit, rat, mouse, human sera (Sigma-Aldrich, 1 ml/pulldown) or a PBS control for 60 min. The beads were rapidly washed with 3× PBS washes in 5 min total. Bound protein was eluted using SDS-polyacrylamide gel electrophoresis (PAGE) sample buffer and heating at 50 °C.

**Protein electrophoresis and visualization**. Protein samples were prepared and analysed using by SDS-PAGE and standard methods. Western blotting used Immobilon-P membrane (Millipore) and standard methods. Rabbit FHR antiserum was generated against recombinant FHR (Covalab) and used as primary antibody (CUK-1457 1, 1:8000 dilution). A mouse moAb (L13D6, 1:100 dilution) to a paraflagellar rod protein was used as loading control (kind gift of Keith Gull)[53]. Donkey or goat anti-rabbit and donkey anti-mouse horseradish peroxidases (HRP; 1:5000 dilutions) were used as secondary antibody. Streptavidin HRP (strep POD) was used at varying dilutions as specified.

**MS for protein identification**. Pulldown samples were fractionated by SDS-PAGE (Criterion XT-PreCast Gel, 12 % Bis-Tris, Bio-Rad) and protein bands were excised, trypsin-digested and run on an liquid chromatography-tandem MS (LC-MS/MS) system by electrospray ionization-quadropole-time of flight (ESI-QUAD-TOF) in the Advanced Proteomics Facility (University of Oxford) using standard procedures. Results were searched against the UniProtKB/Swiss-Prot database and analysed by Mascot.

**Hexahistidine and AviTag fusion trypanosome proteins**. For SPR, cross-linking and CD analyses, synthetic genes were synthesized that were optimized for expression in *E. coli*. After cloning into pET15b, proteins encoded a N-terminal hexahistidine (6×his) tag followed by a TEV cleavage site, FHR/mutant 3 residues 25–249, a GSGSGSS linker and a C-terminal AviTag. To generate mutant 2, mutagenesis was performed on the vector containing the WT FHR sequence with the QuikChange II Site-Directed Mutagenesis Kit (Aligent) using primers 5′-AA TGATAATCTGGAAGCAGCACTGGAAGCAACCAAAGCACTGTGTGAA-3′ and 5′-TTCACACAGTGCTTTGGTTGCTTCCAGTGCTGCTTCCAGATT ATCATT-3′.

For crystallography, FHR residues 25–251 were cloned from TREU 927 genomic DNA into a pOPINF vector (Oxford Protein Production Facility) containing an N-terminal 6×his tag and a PreScission cut site, which leaves an N-terminal glycine–proline post cleavage. Proteins were expressed in *E. coli* BL21 (DE3) in the same manner as described for GST fusion proteins and purified with nickel-nitrilotriacetic acid (Ni$^{2+}$-NTA) affinity chromatography. Purified proteins were buffer exchanged into PBS. His tags were removed by incubation for 2 h with TEV protease at 37 °C or PreScission protease at room temperature[54], which were subsequently removed with another round of Ni$^{2+}$-NTA affinity chromatography. Further processing of proteins was performed as required by the assay and is described in the following sections.

**Cloning, expression and purification of mammalian proteins**. Sequences encoding bovine FH (Q28085, UnitProt) and various truncations were synthesized after codon optimization for expression in Chinese hamster ovary (CHO) cells: complete (residues 18–1236), domains 4-5-6 (residues 207–385), domains 7-8-9 (residues 385–564) and domains 18-19-20 (residues 1052–1236). Sequences containing mouse domains 4-5-6 (P06909, UnitProt, residues 207–387) and human (P08603, UnitProt, residues 207–387) were also synthesized in the same manner. All of the above were cloned into a pDest12 vector for mammalian expression driven by a cytomegalovirus promoter. The CD33 signal peptide was used for protein secretion and tags were placed at either the N or C terminus based on the composition of the protein. For example, tags were not placed at the C terminus of proteins containing FH domain 20, as this is the native C terminus. The final expression vector encoded the CD33 signal peptide, followed by either (1) an N-terminal 10 × his tag, an AviTag, a GSGSGS linker before the ORF (for domains 18-19-20) and also a TEV site before the ORF (for full-length bovine FH), or (2) the ORF, a C-terminal GSGSGS linker, AviTag and 10×his tag (for domains 4-5-6 and 7-8-9). Further truncations were generated by PCR of the vector encoding bovine FH. Domain 5 (residues 264–323) was amplified with primers 5′-GCTAG CGGATCCTGGCAGCGGTAGCGAGAACCTGTACTTTCAAGGCAGCGGCGAG ATCACATGCGACCCTCC-3′ and 5′-GCTAGCTCATCACTTCCAGGCGCATC TAGGC-3′ for N-terminal tags, a TEV site and an extra serine–glycine linker.

The expression plasmids were used to transfect G22 CHO cells, which were grown in 500 ml serum-free CCM8 medium (SAFC). Cells were subsequently cultured for 8 days and supplemented on days 1, 3 and 6 by the addition of 3.3% F9 and 0.2% F10 cell feed (AstraZeneca). Secreted recombinant protein was recovered from the culture supernatant by Ni$^{2+}$-NTA affinity chromatography. Purified proteins were buffer exchanged into PBS and further processing of proteins was performed as below.

**Biotinylation**. Biotinylation of proteins containing an AviTag was performed, by mixing protein at 30 µM with 0.5 µM BirA, 5000 µM ATP and 300 µM biotin. They were incubated at room temperature overnight, before desalting to remove excess biotin. Biotinylation was assessed by western blotting using strep POD.

**Purification of bovine FH from serum**. Recombinant FHR (7 mg) was immobilized onto 8 ml *N*-hydroxysuccinimide-activated agarose spin columns. The FHR-agarose was incubated with adult bovine serum. Flow-through and subsequent washes with PBS were discarded and bovine FH was eluted with 0.1 M glycine-HCl pH 2.5. Elution fractions were immediately neutralized with 1 M phosphate buffer pH 8.5. Elution fractions were dialysed against PBS.

**Surface plasmon resonance**. Biacore T-100 and T-200 (GE Healthcare) were used to perform SPR analyses of protein–protein interactions. All analyses were performed at 20 °C in 10 mM HEPES, 150 mM sodium chloride, and 0.005% Tween-20 pH 7.5. Biotinylated FHR or mutants were immobilized on a streptavidin-coated chip (Series S Sensor Chip SA). Ligand was serially diluted twofold into running buffer, flowed over the chip surface, followed by running buffer. Regeneration used 0.1 M glycine-HCl pH 2.5. Binding responses were obtained by subtracting a blank response from the FHR/mutant responses. A two-state reaction model (bovine FH) or a 1 : 1 interaction model (bovine FH domains 4-5-6 and 5) were fit to blank subtracted kinetic sensorgrams using the BIAevaluation software, enabling the determination of the affinity ($K_D$) and kinetics of the interaction ($k_{on}$, $k_{off}$). For interactions with a fast dissociation rate ($k_{off}$) (mouse and human FH

domains 4-5-6), steady-state levels as a function of ligand concentration were analysed to obtain $K_D$ only.

**Mouse plasma for SPR.** $Cfh^{+/+}$ and $Cfh^{-/-}$ mouse plasma were obtained as a kind gift from The Sanger Mouse Genetics Programme (Sanger MGP). For each $Cfh^{+/+}$ and $Cfh^{-/-}$ sample, plasma from three age-matched female mice were pooled (full strain C57BL/6 N(25%) and C57BL/6NTac(75%)).

**Chemical cross-linking and MS.** Cross-linking experiments were performed with DSS H12 and D12 (Creative Molecules). FHR/mutant 3 and purified native bovine FH were incubated together in PBS or individually with PBS as a control. All tubes contained the same molarities of components (10 µM FHR/mutant 3, 3.3 µM FH). A titration of cross-linker was added to aliquots of the protein solutions for 20 min. Samples were analysed by SDS-PAGE and western blotting. For MS analysis, biotinylated FHR/mutant 3 were incubated with bovine FH. Half of the mixture was incubated with 125 µM H12 DSS and half with 125 µM D12 DSS. The reaction was stopped using a final concentration of 10 mM methylamine pH 8 and samples were treated with PNGase F under native conditions. The sample containing FHR + FH + D12 DSS was mixed in equal volume with mutant 3 + FH + H12 DSS and the sample containing FHR + FH + H12 DSS was mixed in equal volume with mutant 3 + FH + D12 DSS. Samples were fractionated by SDS-PAGE (4–12% Bis-Tris NuPAGE gel, Invitrogen) and high-molecular-weight cross-linked proteins were excised.

Thiols were reduced with tris(2-carboxyethyl)phosphine and alkylated with chloroacetamide before overnight digestion with Sequencing Grade Modified Trypsin (Promega) at 37 °C. Peptides were separated on an EASY-nLC 1000 UHPLC system (Proxeon) and electrosprayed directly into a Q Exactive Hybrid Quadrapole Orbitrap LC-MS/MS mass spectrometer. Peptides were trapped on a C18 PepMap 100 pre-column (300 µm × 5 mm, 100 Å) using 0.1% formic acid in water at a pressure of 500 bar and then separated on an in-house packed analytical column (75 µm × 50 cm packed with ReproSil-Pur 120 C18-AQ, 1.9 µm, 120 Å, Dr Maisch GmbH) with a linear gradient (10–55% of 0.1% formic acid in acetonitrile for 45 min, flow rate = 200 nL/min). Full-scan MS spectra were acquired in the Orbitrap (scan range 350–2000 m/z, resolution 70,000, AGC target $3^6$ ions, maximum injection time 100 ms). After the MS scans, the ten most intense peaks were selected for higher-energy collisional dissociation (HCD) fragmentation at 30% of normalized collision energy. HCD spectra were also acquired in the Orbitrap (resolution 17500, AGC target $5^4$ ions, maximum injection time 120 ms) with first fixed mass at 100 m/z. Charge states 1+ and 2+ were excluded from HCD fragmentation.

MS data were converted into mgf format using pParse and searched using the pLink software[55]. The database contained the target proteins only. Search parameters were as follows: maximum number of missed cleavages = 2, fixed modification = carbamidomethyl-Cys, variable modification 1 = Oxidation-Met, variable modification 2 = Glu to pyro-Glu. Data were initially filtered by $E$-value < $1.0^{-8}$. Cross-links were further filtered/inspected with specific emphasis on fragmentation patterns.

**CD and thermal melts.** Far ultraviolet CD spectroscopy was performed on WT, mutant 2 and mutant 3 FHR using a J-815 Spectropolarimeter (Jasco) and a Peltier temperature control unit. Proteins were analysed at 0.2–0.25 mg/ml in 50 mM sodium phosphate pH 7.5 with a 1 mm path cell between 190 and 260 nm at 20 °C. For each protein, ten runs were acquired at a scanning speed of 50 nm/min with the buffer baseline subtracted. For thermal melts, proteins were analysed at 0.05–0.09 mg/ml in 50 mM sodium phosphate pH 7.5 with a 1 mm path cell between 190 and 250 nm at 0.5 °C increments from 20 °C to 96 °C.

**Crystallography.** His tags were removed from the FHR and domain 5 by Pre-Scission and TEV proteases as detailed above, followed by Ni$^{2+}$-NTA affinity chromatography and dialysis into 20 mM Tris, 150 mM NaCl pH 8. After concentration (Amicon Ultra centrifugal filters, 10,000 or 3,000 Da molecular weight cutoff), proteins were mixed in a 1 : 1.05 molar ratio with domain 5 in slight excess. The mixture was purified using size-exclusion chromatography with a Superdex 200 16/60 column (GE Healthcare) into 10 mM Tris, 50 mM NaCl pH 8. Purified complex was concentrated to 14 mg/ml and subjected to sitting drop vapour diffusion crystallization. A mosquito LCP (TTP Labtech) was used to mix 100 nl of complex with 100 nl reservoir solution from Swissci 96-well commercially available screens (Molecular Dimensions). Crystals were obtained at 18 °C from a JCSG+ plate in a well solution (H6) of 0.1 M ammonium acetate, 17% (w/v) polyethylene glycol 10,000, 0.1 M Bis-(2-hydroxyethyl)imino-tris(hydroxymethyl)methane pH 5.5. Crystals were cryoprotected by transfer into well solution augmented with 25% (v/v) glycerol and cryocooled by transfer to liquid nitrogen.

Data were collected at the Diamond Light Source on beamline I03 and crystals diffracting to 2.70 Å were used for structure determination and processed using XDS[56] and Aimless from the CCP4 suite[57]. Molecular replacement was performed using Phaser[58] with a truncated version of the crystal structure of TcHpHbR (PDB 4E40) as an alanine search model. This process placed two copies of the FHR in the asymmetric unit. After initial building by hand in Coot[59] and refinement in BUSTER[60], clear density was present for bovine FH domain 5 near the N terminus

of both copies of the FHR in the asymmetric unit, allowing subsequent rounds of model building in Coot and refinement in BUSTER. Generation of a model of FHR with FH and C3b was performed in Coot.

**_T. brucei_ PCF cell culture.** _T. brucei_ PCF strain J10[61] was cultured in supplemented differentiating Trypanosome medium[62] containing 10% (v/v) heat-inactivated fetal calf serum (HI FCS) and 6.4 µg/ml haemin. Penicillin (86 U/ml) and streptomycin (86 µg/ml) were added. Cells were maintained in suspension culture at 27 °C in a humidified incubator with 5% $CO_2$ between $1 \times 10^6$ and $1 \times 10^7$ cells/ml, and all experiments and transfections were performed when the cells were in mid-log phase growth of $3–8 \times 10^6$ cells/ml.

**Generation and confirmation of PCF FHR$^{-/-}$ cell lines.** A strategy using long oligonucleotide primers was used to delete the gene for the FHR on both alleles in PCFs[63]. In brief, forward primers contained 80 base pairs upstream of the start codon of the FHR gene and 20 base pairs upstream of the start codon for a blasticidin or neomycin (G418) resistance gene. Reverse primers contained the last 20 base pairs of the resistance gene including the stop codon and around 80 base pairs downstream of the stop codon of the FHR gene. The gene flanking sequences were obtained from the TREU 927 reference genome[49]. PCR products were introduced into PCF _T. brucei_ by standard transfection procedure[64] and transformants were selected with 10 µg/ml of blasticidin and 45 µg/ml of neomycin.

To confirm deletion of the FHR gene, genomic DNA was extracted from WT and suspected FHR knockouts. Genomic DNA (1 µg) was digested with NdeI and NcoI followed by a Southern blot analysis and standard methods. DNA probes prepared by PCR covered the entire FHR ORF of 816 base pairs with 5′-ATGAT GATTTCCCGCGCTTTG and 5′-GAAAGAGAGGGCCGTGGCGGC-3′ or a region 3′ of the ORF of 1023 base pairs with 5′-GCTCGCAGCTAATGATGATCC and 5′-CGTGATGCCACGAGTCCCTTC-3′.

**Tsetse fly infections with PCFs from an in vitro culture.** Male and female tsetse flies, _Glossina morsitans morsitans_ and _Glossina pallidipes_, were infected with WT and three independent FHR$^{-/-}$ clones as PCFs from culture[61]. Tsetse flies were maintained on washed horse red blood cells[61] and thus in the absence of active complement. Infection of the midgut was assessed by dissection after 6 days. Infections were scored as positive if live PCFs were observed on microscopic examination[65]. As both WT and FHR$^{-/-}$ infections were observed, flies were maintained for 28 days to allow progression to the salivary glands (see below). For immunofluorescence assays, female _G. pallidipes_ flies were infected with WT and one FHR$^{-/-}$ clone, and were maintained in the same manner as describe above, and the midgut (day 3) and midgut and proventriculus (day 10) were dissected and subjected to immunofluorescence assays for determination of FHR expression.

**Generation and maintenance of BSF FHR$^{-/-}$ cell lines.** Infected salivary glands from tsetse flies with WT and two FHR$^{-/-}$ clones were used to infect BALB/c mice[61]. The experiments designed for this study were carried out according to the UK Animals (Scientific) Procedures Act under a licence (30/3046) from the UK Home Office and approved by the University of Bristol ethics committee. To adapt BSFs to culture, aliquots of infected mouse blood were transferred into modified HMI-11 media[66] with 1% (w/v) methylcellulose, 10% HI FCS[67] and cultured at 37 °C with 5% $CO_2$. Cells were maintained between $1 \times 10^5$ to $1 \times 10^6$ cells/ml and all experiments were performed when the cells were in mid-log phase growth of $5 \times 10^5$ to $1 \times 10^6$ cells/ml. BSF cells were adapted to standard culture by twofold dilution every 2–3 days or more regularly as necessary, along with an incremental decrease in methylcellulose.

The induction of stumpy BSFs in culture was performed by following an established protocol[42,43]; 8-(4-Chlorophenylthio)adenosine 3′,5′-cyclic monophosphate sodium salt (pCPT-cAMP, Sigma) was added to cultured slender BSFs (100 µM) and cells were collected or analysed 24 h later.

**Live-cell-binding assays with FH and moAb.** PCFs were incubated in culture with 100 nM of fluorescently labelled FH purified from bovine serum (see above) or moAb raised against the FHR by phage display (AstraZeneca)[68,69]. In brief, phage display selection strategy involved panning selections with biotinylated FHR on streptavidin-coated plates or with unlabelled FHR on MaxiSorp plates, and soluble selections with biotinylated FHR on magnetic streptavidin-coated Dynabeads. GST-tagged and -untagged FHRs (see above) were used throughout and were either left as is or biotinylated. When a GST-tagged protein was used, a GST de-selection step was performed as a control. In all selections, $1 \times 10^{12}$ phage particles were incubated with immobilized FHR, followed by extensive wash steps (PBS or with 0.1% Tween-20), de-selections (e.g., removal of non-antigen-binding phage-scFv by pre-incubation with beads) and elution of bound phage with trypsin[69].

_E. coli_ TG1 were infected with eluted phage, followed by rescue with super-infection of M13 KO7 helper phage. These phage were used for one to two further rounds of selection. The enriched anti-FHR phage-scFv outputs were sequenced to assess diversity and then tested further for FHR binding using phage enzyme-linked immunosorbent assay. Bovine serum albumin and GST were negative control antigens and phage displaying a negative control scFv. Binding of phage-scFv was assessed with an anti-M13 HRP-conjugated antibody. Ten unique

phage-scFv were obtained as soluble fragments in periplasmic extracts[68] and were selected for further binding analysis, and ranked using homogeneous time-resolved fluorescence and BioLayer Interferometry on an Octet RED384 by standard procedures. Nine purified anti-FHR scFv were selected, which bound unique epitopes and converted into full-length human IgG1 using standard techniques. Secreted antibodies were purified using protein A affinity chromatography and labelled using an AlexaFluor 594 NHS ester labelling kit (ThermoFisher)[69]. After 2 h of incubation, live cells were fixed in 1% formaldehyde for 10 min. The cells were washed, resuspended in PBS and loaded onto poly-lysine slides for visualization by microscopy. All nine anti-FHR moAbs were tested and one is shown in this work.

**Mouse infections with BSFs.** Stabilates of infected mouse blood with *T. brucei* BSF J10 WT and two FHR$^{-/-}$ clones were thawed and passaged via intraperitoneal injection into immunosuppressed (Endoxan 300 mg/kg) BALB/c mice (BALB/cByJ, Charles River)[65]. Immunosuppression was performed on the day of inoculation. On day 3 post infection, blood from infected mice was diluted in PSG and was injected into a further set of immunosuppressed BALB/c mice ($5 \times 10^6$ cells per mouse, 5 mice per cell line). On day 4 post infection, blood from infected mice was diluted in phosphate buffered saline glucose (PSG) and injected into a further set of immunocompetent BALB/c mice ($1 \times 10^6$ cells per mouse, 5 mice per cell line). Parasitaemia was monitored by microscopy of whole blood[70]. All mice were female and age-matched. The experiments designed for this study were approved by the regional Ethic Committee for Animal Experimentation CEEA-LR 36 under project number 2018012915201897v2 and authorized by MENESR (French Ministry for Higher Education and Research).

**Mathematical modelling of BSF infections in mice.** Mathematical modelling of *T. brucei* infection dynamics in immunocompetent mice[45,46] has been performed here with minor modification. The model, as first described[46], was fitted to the parasitaemia of individual immunocompetent mice in this study, but the number of antigenic variants was limited to three and a variant 3-induced immune response was removed. PAD1 expression and slender, intermediate and stumpy cell proportions were not measured and therefore not fitted. However, the dynamics of slender and stumpy forms were modelled. The full details of the model are available in Supplementary Data 1.

**Tsetse fly infections with BSFs from infected mice.** Female *Glossina palpalis gambiensis* were fed on the bellies of infected, anaesthetized mice[65] and subsequently on sheep blood by in vitro membrane feeding. Feeding on mice was performed seven times and each time parasitaemia was matched closely (less than twofold difference). The midguts from starved tsetse flies were dissected from 6 to 16 days post infection. Midgut infections were scored as positive if live PCFs were observed on microscopic examination[65]. The experiments designed for this study were approved by the regional Ethic Committee for Animal Experimentation CEEA-LR 36 under project number 2018012915201897v2 and authorized by MENESR (French Ministry for Higher Education and Research).

**Immunofluorescence assays.** WT and FHR$^{-/-}$ BSF cells were diluted to $1 \times 10^5$ cells/ml or $4 \times 10^5$ cells/ml + pCPT-cAMP as described above. After 24 h, $1 \times 10^7$ cells were collected and washed with serum-free HMI-11, with or without pCPT-cAMP. After resuspension in Voorheis-modified PBS (vPBS = PBS supplemented with 46 mM sucrose, 10 mM glucose pH 7.6), cells were fixed for 30 min by adding an equal volume of 8% paraformaldehyde. After dilution with vPBS, cells were collected by centrifugation and settled onto poly-lysine slides in vPBS. Slides were washed with vPBS, 10 mM ethanolamine pH 8 in vPBS, vPBS ± 0.1 % Triton X-100 and vPBS[44]. After blocking for 1 h in 5% donkey serum in vPBS, slides were incubated with primary antibody for 1 h and then washed with vPBS[44]. Slides were then incubated with secondary antibody for 1 h and washed again[44]. Slides were incubated with Hoechst 33258 (0.1 mg/ml) for 5 min, washed and mounted with FluorSave reagent[44]. The primary antibody used here was rabbit FHR antiserum described above (1 : 200, 1 : 400 and 1 : 800 dilutions) and secondary antibody was donkey anti-rabbit AlexaFluor 488 (1 : 1000 dilution). Microscopy was performed using a Zeiss Axioimager M1; images were recorded using the Axiovision software (Zeiss) and the same settings were used for all samples within a set of experiments. Images were imported into Adobe Photoshop for figure preparation. The midgut and proventriculus from tsetse flies infected with WT and FHR$^{-/-}$ PCF cells as described above were dissected[61]. The organs were transferred directly to slides in vPBS and fixed in an equal volume of 4% or 8% paraformaldehyde. The remaining steps of the IFA were as described for cultured BSF cells.

**Reporting summary.** Further information on research design is available in the Nature Research Reporting Summary linked to this article.

## Data availability

The Source Data file contains the source data underlying all the figures as stated. The crystal structure has been deposited in the Protein Data Bank (PDB ID 6XZ6) and all other data are available from the authors on request. Correspondence and requests for materials should be addressed to M.K.H. (matthew.higgins@bioch.ox.ac.uk) and M.C. (mc115@cam.ac.uk).

## Code availability

The code for the mathematical model in this work is deposited in GitHub and the links and detailed description of the model can be found in Supplementary Data 1.

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

## Acknowledgements

This work was funded by MRC grant MR/P001424/1 to M.C. and M.K.H., and Magdalene College, University of Cambridge, scholarship to O.J.S.M. This project has received resources funded by the European Union's Horizon 2020 research and innovation programme under grant agreement number 731060 (Infravec2) to O.J.S.M., Labex Parafrap ANR-11-LABX-0024 to J.-M.B and BBSRC grant BB/M008924/1 to W.G. M.C. (217138/Z/19/Z) and M.K.H. (101020/Z/13/Z) are Wellcome Trust Investigators. We thank Ed Lowe and the beamline scientists at I03 at Diamond Light Source; the Oxford Protein Production Facility (OPPF); Federico Rojas and Keith Matthews, University of Edinburgh, for stumpy trypanosome lysate; Raymond Stevens for PreScission expression plasmid (NIH grant P50 GM073197); Katherine Stott and the BioPhysics Facility, University of Cambridge; the Advanced Proteomics Facility, University of Oxford; InterTryp IRD-CIRAD and Vectopole Sud, Montpellier, France insect facilities; Andrew Herbert and Paul Barlow, University of Edinburgh for advice and discussions; and The Sanger Mouse Genetics Programme (Sanger MGP) for the kind gift of reagents.

## Author contributions

M.C. conceived and O.J.S.M., M.K.H. and M.C. designed and managed this study. O.J.S.M., P.M., M.K.H. and M.C. interpreted and analysed data throughout. O.J.S.M. conducted all investigations and experiments, with specific contributions hereafter. J.-M.B., L.P., S. Ravel, W.G. and M.C.T. performed and designed tsetse fly and animal experiments. N.J.S. modelled trypanosome infections. J.D.S., S.H. and S.M. prepared and analysed mass spectrometry data. C.T. aided reagent production. S. Rust., T.J.V. and R.M. managed monoclonal antibody production. O.J.S.M. and M.K.H. prepared crystals and solved the structure. O.J.S.M., M.K.H. and M.C. wrote the manuscript, with critical revisions by P.M. and input from all authors.

## Competing interests

The authors declare no competing interests.
