## [Peer Review File · Nature Communications]

Reviewers' Comments:

Reviewer #1:

Remarks to the Author:

In this paper, the authors identify and characterize a receptor for the complement regulator factor H in the African trypanosome *T. brucei*. They present a model of mode of action based on a number of different experimental approaches such as structural analysis, interaction analyses between the receptor and its ligand, differential expression in different life cycle stages of the trypanosome and infection studies of mice and tsetse flies. Contrary to intuition, they find that expression of the receptor, which binds Factor H, a regulator of complement activation in the mammalian host, is not detectable in the slender bloodstream form, but is upregulated in stumpy BSF, and the tsetse resident procyclic and mesocyclic forms which come into contact with mammalian blood during feeding, and is again absent in epimastigotes in the tsetse fly, which do not come into direct contact with the blood meal. From infection studies of mice and tsetse flies with wild type (FHR+/+) and FHR-/- trypanosomes the authors conclude that the presence of the receptor increases transmission of trypanosomes from the mammalian host to tsetse flies by increasing the stumpy population in the infected mammal as modelled from infections studies on mice. This is an interesting concept and advances our knowledge of the sophisticated mechanisms *T. brucei* has evolved to survive and thrive in its host and vector.

Minor points

- 1) Please clarify the following sentence on p. 6, l. 1-3: 'This analysis resolved which of the two FHR and two FH domain 5 copies present in the asymmetric unit of the crystal comprised the physiological complex.' Sorry if I missed this, but which one comprised the physiological complex and why?
- 2) On page 7 the authors analyse expression of the FHR in tsetse fly trypanosome stages following infection of the vector with parental FHR+/+, showing that FHR expression occurs in procyclic and mesocyclic forms, but not epimastigote forms. A control experiment with FHR-/- cells was performed, which obviously does not express FHR, but no further information about this experiment is given other than that tsetse flies were infected with this. In the absence of complement I assume FHR+/+ and -/- will have infected equally well, but this should be stated.
- 3) p.9: 'FHR is predicted to reduce stumpy BSF lifespan in mice' should read increase instead of reduce
- 4) The manuscript should be carefully proofed to eliminate a number of errors in the text (spelling, repeats and missing words).

Reviewer #2:

Remarks to the Author:

African trypanosomes are extracellular parasites which must escape the effects of the complement system, both while replicating in the mammalian bloodstream, as when they are present in the bloodmeal of the tsetse fly insect vector. Despite its great importance, very little is known about how this trypanosome evasion of the complement system occurs.

This study identifies a trypanosome factor H receptor (FHR) which "catches" the host (complement inhibitory) factor H, thereby facilitating trypanosome complement evasion. This mechanism of complement evasion is used by other pathogens. However to the best of my knowledge, this would be the first well documented example of this occurring in a unicellular parasite.

It is quite a fantastic discovery that the authors have made. The data appear very solid with extensive controls. The authors provide structural data, as well as extensive validation of protein interactions. The manuscript is clearly written, and the data nicely presented. I think that it will be found very interesting by the broader readership.

Interestingly, this *T. brucei* FHR receptor does not appear to play an important role in slender replicating bloodstream form trypanosomes. In contrast, FHR is expressed in the quiescent "stumpy" form trypanosomes, as well as trypanosome life-cycle stages present in the tsetse fly insect vector. Here, trypanosomes are exposed to blood complement, but do not have the protective VSG coat present in the bloodstream form, making them susceptible to lysis.

I thought that it was a very interesting and important manuscript. However, I think that it could be improved if the authors discussed the following issues in a bit more detail to improve comprehension of the implications of the findings for the general reader.

1. What are the authors' thoughts regarding the reduced life expectancy of the FHR- stumpy trypanosomes in mice? This is particularly as the stumpy form is thought to be more resistant to the effects of complement. Does this resistance need to be re-evaluated, and are the stumpy trypanosomes in fact getting lysed?
2. If they are getting lysed, what is the evidence that the mouse strain used in these experiments has a lytic complement system? Many lab mouse strains are deficient for complement C5, and can not form a MAC.
3. Why would stumpy form trypanosomes need FHR but not the slender forms? Alternatively, is the idea that the stumpy stage has reduced rates of VSG recycling and therefore can not 'clean' their VSG surface as effectively as slender form trypanosomes? This could result in increased phagocytosis by macrophages. Spelling out various outcomes would be useful.
4. *Trypanosoma brucei* naturally infects a large range of mammalian species in Africa. Could the authors say something about protein sequence diversity of the different factor H molecules in different mammals? How do the authors think that a single trypanosome FHR receptor can recognise them all? The authors describe a protein structure with two helices which have splayed apart forming a binding pocket which could accommodate this, (p. 11). However, this could be further elaborated on with regards to referring to factor H variants present in different species of mammal.

Reviewer #3:

Remarks to the Author:

This is a fabulous manuscript: the structural studies, the involvements of murine and insect models, and the conceptual exploration of a novel microbial mechanism of immune evasion. The field has long been curious as to how *T. brucei* parasites evade the activity of complement, or rather the mechanistic reason why certain life cycle stages are more resistant than others. This discovery not only addresses that gap in knowledge, but simultaneously adds a new pathogen that specifically utilizes the host Factor H as an evasion mechanism. Strikingly, all 3 organisms with a molecular characterization of their Factor H binding receptors (now 4 including *T. brucei*) bind Factor H in structurally very distinct ways, thus suggesting a model of convergent evolution whereby numerous pathogens have independently developed a similar mechanistic complement evasion strategy. Additionally, the manuscript is professionally prepared with elegant figures and a logical flow of text, which reads very easily.

Major comments

1) There is no actual experimental evidence that the putative *Trypanosoma brucei brucei* Factor H receptor binds to Factor H in vivo or ex vivo. The ability of the receptor to interact with its putative target in an SPR experiment, crosslinking experiment and the fact that the receptor co-crystallizes with one domain of the target both suggest that they can bind together when in close proximity at high concentrations.

"FHR/mutant 3 and purified native bovine FH were incubated together for 30 min at room temperature in PBS or individually with PBS as a control. All tubes contained the same molarities of components (10 μ M FHR/mutant 3, 3.3 μ M FH)."

"The interaction was confirmed using surface plasmon resonance (SPR); the putative receptor was expressed, biotinylated at a single site near the C-terminus and immobilised on a streptavidin chip. Tb927.5.4020 bound bovine FH, either purified from serum or produced as a recombinant protein, with a K_D \sim 150 nM (Fig. 1b and Supplementary Fig. 1b-f)."

What does this interaction look like on a trypanosome? We don't know. The authors seem to have attempted to address this potential pitfall with the experiment presented in Fig 5B, wherein it is shown that a monoclonal antibody binds to the cell surface of cultured trypanosomes.

"A FHR moAb was used so that the cells could be kept in culture medium during the incubation with ligand; any labelled FH added to medium would be diluted by FH in the serum in the culture medium."

"T. brucei PCF strain J1067 was cultured in Differentiating Trypanosome Medium (DTM)68 containing 10 % (v/v) heat inactivated foetal calf serum (HI FCS) and 6.4 μ g/ml hemin69."

"PCFs were incubated in culture with 100 nM of a fluorescently labeled moAb raised against the FHR by phage display as performed previously."

The affinity of this antibody for the receptor is not stated but 100nM was sufficient for detection of binding (one would add 5-10 fold above the K_D to saturate the receptor). The affinity for the trypanosome FHR for FH is 150nM by SPR, this should be possible to visualize even in the 10% HI FCS. What is the concentration of FH in FCS?

What region of the receptor does the antibody bind? It's possible that the antibody just binds with very high affinity to a highly exposed region of the receptor. Does the antibody out compete FH binding to FHR either directly or by steric hindrance?

Additionally, the authors report that a "fraction" of the antibody was endocytosed. Have the authors considered that bound Factor H is being rapidly internalized. Factor H is just slightly larger (155 kDa) than an IgG (150 kDa), thus at least suggesting that it could possibly be internalized via the same hydrodynamic drag mechanism. If so are other complement components also internalized?

To address this issue, a number of possible experiments could be done. Using immunofluorescence to show that Factor H can be detected at the surface of the cells (but not the knockout cells) would be ideal. I have to assume this was tried with no success. Have the authors tried a Proximity Ligation Assay? Perhaps using the FHR moAb and tagged FH (or FH domains) and FHR^{-/-} cells as the negative control would work.

2) I will acknowledge that my ability to interpret these mathematical models that supposedly support the data is limited. I do not know why the authors did not use PAD1 as a marker of short stumpties, please clarify. Nevertheless, in order to suggest that these knockout lines are attenuated in mice and during transmission due to a susceptibility to complement, why not utilize any of the number of murine models that have disrupted complement pathways? The differences in parasitemia and stumpy life duration should disappear if the same experiments were performed using the C5^{-/-} mice from

Jackson, for example (B10.D2-Hc0 H2d H2-T18c/oSnJ).

Lastly, the simplest possible experiment to do would be to isolate stumpy form parasites from both the WT and knockout lines and do in vitro complement lysis experiments with murine serum derived from WT mice as well as complement inactivated (ideally genetically) mice. At least this would be a financially more feasible option.

Minor comments

1) I am curious as to what other molecules from Bovine serum will bind to this receptor. There are at least a couple more unique bands (albeit with lower abundance) present in figure 1A. The authors have indicated that they performed mass spec on the bands. Can we see the raw data from that experiment in the supplementary materials?

2) Page 9, lines 25 and 26. Fix sentence.

3) Page 10, line 15: "reduced but did NOT abolish" is what I believe is intended.

4) Conceptually, it is surprising that the receptor is not expressed by slender BSFs. I wonder if the authors can expand on this in the discussion section. They currently have one or two sentences on the topic, wherein they suggest that the hydrodynamic clearance of complement might be sufficient for slender forms to evade lysis.

"How do slender BSFs protect themselves against the alternative complement pathway with little or no FHR expression? The process of hydrodynamic flow coupled to endocytosis to rapidly clear surface-bound antibodies 25 24 may be sufficient in slender BSFs to remove deposited C3b or C3bBb before downstream assembly of MAC occurs. It is also probable that there are additional mechanisms to counteract complement."

However, the 2007 Engstler et al., paper also shows that stumpies clear IgG from their surface approximately twice as fast as slenders if I'm not mistaken. This would suggest that the slenders are actually less efficient at clearing surface bound immune complexes such as deposited C3b.

Reviewer #4:

Remarks to the Author:

Macleod et al. report the identification of a *Trypanosoma brucei* receptor for factor H (fH), a downregulator of the complement system. The authors demonstrate interaction between this trypanosomal receptor, called FHR, and bovine fH by pulldown and crosslinking experiments using bovine serum, and direct interaction between the purified forms of these proteins by surface plasmon resonance. The latter shows that the interaction between FHR and fH is quite tight, with a KD of 150 nM. The authors also identify several FHR amino acids as candidates for fH-interaction, and show that some of these are essential for fH-binding; a mutant FHR protein containing substitutions at these amino acids provides a valuable negative control. The authors also identify fH domain 5 as sufficient for interaction with FHR, and determine the X-ray crystal structure of fH domain 5 bound to FHR. Structural modeling of the intact fH-FHR complex suggests that fH remains free to interact with and downregulate the C3 convertase. However, whether this indeed occurs is not tested (Major Point #1). The authors move from these in vitro studies to several in vivo ones. For the latter, they construct several independent FHR^{-/-} *T. brucei* lines. The authors determine that FHR is expressed, as detected by antibodies, in the stumpy blood stream form but not in the slender blood stream form, which developmentally precedes the stumpy form. The greatest expression of FHR, however, occurs in the procyclic form, which occurs in the midgut of tsetse flies. It wanes in the mesocyclic form, which migrate from the midgut to the proventriculus of tsetse flies, and is not detectable in the epimastigote form, which migrate from the proventriculus to the salivary gland of tsetse flies. The authors also

show that FHR is accessible on the surface of the procyclic form using a monoclonal antibody. These results suggest a role for FHR in transmission between infected mammalian hosts and tsetse flies, which the authors confirm through mouse infection experiments. However, the authors do not confirm that FHR interacts with murine fH domain 5, as opposed to bovine fH domain 5 as in the in vitro experiments (Major Point #2). Mild differences are seen in parasitemia caused by wild-type vs. FHR^{-/-} *T. brucei*, but based on an apparently well-established mathematical model, the authors draw the conclusion that loss of FHR results in a ~2-fold decrease in the stumpy blood stream form due to a decrease in lifespan of this form from ~27 h to ~7 h. It is remarkable that a mild effect in parasitemia can be modeled into a major change in lifespan (Other Issues #1). Lastly, the authors find that loss of FHR leads to a ~2.5-fold decrease in the number of *T. brucei* transmitted from infected mice into the midgut of tsetse flies. The work is solid overall, but indirect in places where direct evidence should be presented. Several major points could be addressed by the authors to strengthen the work.

Major Points

1. It is surprising that the authors do not directly examine the functional effect of recruitment of fH by FHR. For example, is the amount of C3b lower (or C3dg or C3d greater) on the surface of wild-type vs. FHR^{-/-} *T. brucei*?
2. The in vitro experiments are carried out with bovine fH, while the in vivo experiments are carried out in mice. Therefore, it is crucial for the authors to verify that FHR interacts with murine fH domain 5. (Along these lines, it would also be useful to know whether FHR interacts with human fH domain 5.)
3. A clear test of the role of fH and FHR in transmission would be to examine fH-knockout mice. Is there a reason why this was not done?

Other Issues

1. As general readers will likely have to take the mathematical modeling by faith, the authors should provide some explanation of the model and evidence of its predictive power. They should also show how the other parameters in the model behave between WT and FHR^{-/-} *T. brucei*.
2. A fuller explanation of the choice of Tb927.5.4020 would be helpful. Why was Tb927.5.4020 chosen from all the proteins listed in Table S1? Additionally, why were only GPI-anchored proteins considered? Why weren't transmembrane proteins considered?
3. The CD analysis of FHR mutant proteins establishes that structure is conserved (at whatever temperature the experiment was done, apparently not listed), but does not establish whether the mutant proteins are as stable as wild-type FHR. The authors should do a thermal melt or chaotropic denaturation of mutant and wild-type FHR, as monitored by CD.
4. Why are only approximate affinities given for the surface plasmon resonance analysis? Presumably the data yield a specific value with associated errors. These should be stated, and the fits to the data should be shown.
5. The purity of the various FH domains in Figure 2 should be shown. This is especially important for FH 7-9 and 18-20, as these are negative in the assay.
6. There are a surprisingly large number of waters for a 2.7 Å resolution structure. Might this explain the rather large difference between R-free and R-work? Scant information is given about how the structure was refined.
7. The concentrations of proteins used for GST pulldowns should be detailed.
8. The text could use a good dose of editing. There are a number of ungrammatical constructions and some careless mistakes. Some examples follow, but there are a number of others.
 - a. "The mechanisms by which *T. brucei* counteracts the mammalian adaptive immune response are well characterised, antigenic variation at the population level and rapid clearance of surface-bound immunoglobulin at the individual cell level²⁴⁻²⁶."
 - b. "This work started with an assumption that the interactions between trypanosomes and their hosts are likely to be more extensive than those previously characterised, a screen of the *T. brucei* genome was performed to identify putative receptors based on one or both of two criteria:"

c. It has been shown that C3bBb assembly occurred on the surface of BSFs but did the process did not progress further there there was no formation of a MAC by recruitment of complement proteins C5b-930."

Response to reviewer 1

In this paper, the authors identify and characterize a receptor for the complement regulator factor H in the African trypanosome *T. brucei*. They present a model of mode of action based on a number of different experimental approaches such as structural analysis, interaction analyses between the receptor and its ligand, differential expression in different life cycle stages of the trypanosome and infection studies of mice and tsetse flies. Contrary to intuition, they find that expression of the receptor, which binds Factor H, a regulator of complement activation in the mammalian host, is not detectable in the slender bloodstream form, but is upregulated in stumpy BSF, and the tsetse resident procyclic and mesocyclic forms which come into contact with mammalian blood during feeding, and is again absent in epimastigotes in the tsetse fly, which do not come into direct contact with the blood meal. From infection studies of mice and tsetse flies with wild type (FHR+/+) and FHR-/- trypanosomes the authors conclude that the presence of the receptor increases transmission of trypanosomes from the mammalian host to tsetse flies by increasing the stumpy population in the infected mammal as modelled from infections studies on mice. This is an interesting concept and advances our knowledge of the sophisticated mechanisms *T. brucei* has evolved to survive and thrive in its host and vector.

Thank you to reviewer 1 for your positive reception of our work. We have responded to your minor points below.

Minor points

1) Please clarify the following sentence on p. 6, l. 1-3: 'This analysis resolved which of the two FHR and two FH domain 5 copies present in the asymmetric unit of the crystal comprised the physiological complex.' Sorry if I missed this, but which one comprised the physiological complex and why?

We have clarified this section in the main text on lines 154-161. In brief, the physiological complex is that shown in Fig. 3. Mutations in FHR (E31 and Q34), which lie in the shown interface disrupt binding of FHR to FH without affecting the structure of the receptor (Fig. 3 and Supplementary Figs. 2a-d and 3d-g).

2) On page 7 the authors analyse expression of the FHR in tsetse fly trypanosome stages following infection of the vector with parental FHR+/+, showing that FHR expression occurs in procyclic and mesocyclic forms, but not epimastigote forms. A control experiment with FHR-/- cells was performed, which obviously does not express FHR, but no further information about this experiment is given other than that tsetse flies were infected with this. In the absence of complement I assume FHR+/+ and -/- will have infected equally well, but this should be stated.

Indeed, this experiment was done in the absence of complement and we did not observe any significant difference between FHR+/+ and FHR-/- cell lines. We have clarified this in the main text on lines 238-240 and have added the data to Supplementary Fig. 5e. We have also explained this experiment more clearly in the methods section lines 679-688.

3) p.9: 'FHR is predicted to reduce stumpy BSF lifespan in mice' should read increase instead of reduce

We have made this change.

4) The manuscript should be carefully proofed to eliminate a number of errors in the text (spelling, repeats and missing words).

We have done our best to eliminate further errors throughout the manuscript.

Response to reviewer 2

African trypanosomes are extracellular parasites which must escape the effects of the complement system, both while replicating in the mammalian bloodstream, as when they are present in the bloodmeal of the tsetse fly insect vector. Despite its great importance, very little is known about how this trypanosome evasion of the complement system occurs.

This study identifies a trypanosome factor H receptor (FHR) which “catches” the host (complement inhibitory) factor H, thereby facilitating trypanosome complement evasion. This mechanism of complement evasion is used by other pathogens. However to the best of my knowledge, this would be the first well documented example of this occurring in a unicellular parasite.

It is quite a fantastic discovery that the authors have made. The data appear very solid with extensive controls. The authors provide structural data, as well as extensive validation of protein interactions. The manuscript is clearly written, and the data nicely presented. I think that it will be found very interesting by the broader readership.

Interestingly, this *T. brucei* FHR receptor does not appear to play an important role in slender replicating bloodstream form trypanosomes. In contrast, FHR is expressed in the quiescent “stumpy” form trypanosomes, as well as trypanosome life-cycle stages present in the tsetse fly insect vector. Here, trypanosomes are exposed to blood complement, but do not have the protective VSG coat present in the bloodstream form, making them susceptible to lysis.

I thought that it was a very interesting and important manuscript. However, I think that it could be improved if the authors discussed the following issues in a bit more detail to improve comprehension of the implications of the findings for the general reader.

1. What are the authors' thoughts regarding the reduced life expectancy of the FHR- stumpy trypanosomes in mice? This is particularly as the stumpy form is thought to be more resistant to the effects of complement. Does this resistance need to be re-evaluated, and are the stumpy trypanosomes in fact getting lysed?

2. If they are getting lysed, what is the evidence that the mouse strain used in these experiments has a lytic complement system? Many lab mouse strains are deficient for complement C5, and can not form a MAC.

3. Why would stumpy form trypanosomes need FHR but not the slender forms?

Alternatively, is the idea that the stumpy stage has reduced rates of VSG recycling and therefore can not ‘clean’ their VSG surface as effectively as slender form trypanosomes?

This could result in increased phagocytosis by macrophages. Spelling out various outcomes would be useful.

Thank you to reviewer 2 for your positive comments and for facilitating a deeper discussion of the role of FHR in slender and stumpy BSFs. We have included further explanation in lines 298-336 of the discussion section of the manuscript pursuant to questions 1-3. In summary:

All three complement pathways - classical, lectin, and alternative - lead to 1) lysis by MAC formation, 2) opsonisation facilitated by specific receptors (CR1-4 and CR1g or Fc), and 3) inflammation by the formation of proinflammatory anaphylatoxins. Factor H is only involved in inhibiting the alternative pathway, and other negative regulators are involved in the lectin and classical pathways.

The predicted shorter half-life of FHR-/- stumpy forms is most readily explained by more rapid lysis or opsonisation. This would infer that the FHR-/- stumpy forms have a reduced

ability to counteract the effects of cell-surface deposited C3b when compared to FHR-/slender forms.

Previous work has shown that stumpy BSFs are more resistant than slender BSFs to antibody-dependent lysis by the classical complement pathway (McLintock, Turner and Vickerman, 1993; Engstler *et al.*, 2007). It was found that this probably results from a more rapid clearance of antibodies bound to the VSG surface coat in stumpy BSFs and thus lower steady-state levels of surface-bound antibody, as opposed to any mechanism that specifically inhibits downstream processes (Engstler *et al.*, 2007).

The work here does not contradict these findings. There have been no comprehensive comparisons of resistance to the alternative pathway. It is unknown how the slender and stumpy BSF surface interacts or deals with cell-surface deposited C3b. Only one study has shown that the C3 convertase, C3bBb, is able to assemble on a BSF surface, and that it is restricted at this stage by an unknown mechanism (Devine, Falk and Balber, 1986). Differential susceptibility of stumpy and slender BSFs to complement C3 has not been investigated.

The BALB/c mice (BALB/cByJ, Charles River) used in this experiment contain complement C5 (Radovanovic, Mullick and Gros, 2011). Therefore, the most immediate model to explain our observations is that the subtle differences between the slender and stumpy BSF surfaces result in a differential ability to clear surface-deposited C3, possibly due to the presence of components on stumpy, but not slender BSFs, that are not recycled rapidly through the endocytic pathway.

Why is FHR expressed in stumpy forms or how do slender forms avoid the alternative pathway? There are several possibilities: (1) the slender BSF is able to internalise and inactivate C3b at a rate sufficient to prevent MAC formation in most cells and/or (2) the stumpy BSF surface is more susceptible to C3 deposition and so the FHR has evolved to specifically counter this and/or (3) there is an additional mechanism of inactivation possibly an alternative FHR expressed in slender forms. We favour (3) without discarding (1) and (2) as it is likely that the trypanosome expresses a set of receptors/binding proteins for complement components. Indeed, we have identified a C3 binding protein expressed in slender BSFs that we are currently characterising. In addition, the mechanisms evolved to negate complement are very, very, unlikely to depend on a single receptor/binding protein as this would provide a ready route for a host to evolve a countermeasure.

We have included the main points from this in the discussion (lines 298-336). Please see our comments to Reviewer 3, point 2 for further discussion about mice infections.

References:

- Devine, D. V., Falk, R. J. & Balber, A. E. Restriction of the alternative pathway of human complement by intact *Trypanosoma brucei* subsp. *gambiense*. *Infect. Immun.* **52**, 223–229 (1986).
- Engstler, M. *et al.* Hydrodynamic flow-mediated protein sorting on the cell surface of trypanosomes. *Cell* **131**, 505–515 (2007).
- McLintock, L. M., Turner, C. M. & Vickerman, K. Comparison of the effects of immune killing mechanisms on *Trypanosoma brucei* parasites of slender and stumpy morphology. *Parasite Immunol.* **15**, 475–480 (1993).
- Radovanovic, I, Mullick, A. & Gros, P. Genetic control of susceptibility to infection with *Candida albicans* in mice. *PLoS One* **6**, e18957 (2011).

4. Trypanosoma brucei naturally infects a large range of mammalian species in Africa. Could the authors say something about protein sequence diversity of the different factor H molecules in different mammals? How do the authors think that a single trypanosome FHR

receptor can recognise them all? The authors describe a protein structure with two helices which have splayed apart forming a binding pocket which could accommodate this, (p. 11). However, this could be further elaborated on with regards to referring to factor H variants present in different species of mammal.

This response applies to this question, and also to Reviewer 4, point 2. We have performed additional experiments to address this question, by assessing the degree of sequence identity between different factor H variants and assessing factor H binding using pulldown experiments and surface plasmon resonance measurements. These new data can be found in Supplementary Fig. 4 and Supplementary Table 4, and these findings are discussed in the results section lines 175-197 and also in the discussion section lines 345-354.

Response to reviewer 3

This is a fabulous manuscript: the structural studies, the involvements of murine and insect models, and the conceptual exploration of a novel microbial mechanism of immune evasion. The field has long been curious as to how *T. brucei* parasites evade the activity of complement, or rather the mechanistic reason why certain life cycle stages are more resistant than others. This discovery not only addresses that gap in knowledge, but simultaneously adds a new pathogen that specifically utilizes the host Factor H as an evasion mechanism. Strikingly, all 3 organisms with a molecular characterization of their Factor H binding receptors (now 4 including *T. brucei*) bind Factor H in structurally very distinct ways, thus suggesting a model of convergent evolution whereby numerous pathogens have independently developed a similar mechanistic complement evasion strategy. Additionally, the manuscript is professionally prepared with elegant figures and a logical flow of text, which reads very easily.

Thank you to reviewer 3 for your supportive and constructive feedback. We have responded specifically to your comments below.

Major comments

1) There is no actual experimental evidence that the putative *Trypanosoma brucei brucei* Factor H receptor binds to Factor H in vivo or ex vivo. The ability of the receptor to interact with its putative target in an SPR experiment, crosslinking experiment and the fact that the receptor co-crystallizes with one domain of the target both suggest that they can bind together when in close proximity at high concentrations.

“FHR/mutant 3 and purified native bovine FH were incubated together for 30 min at room temperature in PBS or individually with PBS as a control. All tubes contained the same molarities of components (10 μ M FHR/mutant 3, 3.3 μ M FH).”

“The interaction was confirmed using surface plasmon resonance (SPR); the putative receptor was expressed, biotinylated at a single site near the C-terminus and immobilised on a streptavidin chip. Tb927.5.4020 bound bovine FH, either purified from serum or produced as a recombinant protein, with a K_D ~150 nM (Fig. 1b and Supplementary Fig. 1b-f).”

What does this interaction look like on a trypanosome? We don't know. The authors seem to have attempted to address this potential pitfall with the experiment presented in Fig 5B, wherein it is shown that a monoclonal antibody binds to the cell surface of cultured trypanosomes.

“A FHR moAb was used so that the cells could be kept in culture medium during the incubation with ligand; any labelled FH added to medium would be diluted by FH in the serum in the culture medium.”

“*T. brucei* PCF strain J1067 was cultured in Differentiating Trypanosome Medium (DTM)68

containing 10 % (v/v) heat inactivated foetal calf serum (HI FCS) and 6.4 µg/ml hemin69.”
“PCFs were incubated in culture with 100 nM of a fluorescently labeled moAb raised against the FHR by phage display as performed previously.”

The affinity of this antibody for the receptor is not stated but 100nM was sufficient for detection of binding (one would add 5-10 fold above the KD to saturate the receptor). The affinity for the trypanosome FHR for FH is 150nM by SPR, this should be possible to visualize even in the 10% HI FCS. What is the concentration of FH in FCS?

What region of the receptor does the antibody bind? It's possible that the antibody just binds with very high affinity to a highly exposed region of the receptor. Does the antibody out compete FH binding to FHR either directly or by steric hindrance?

Additionally, the authors report that a “fraction” of the antibody was endocytosed. Have the authors considered that bound Factor H is being rapidly internalized. Factor H is just slightly larger (155 kDa) than an IgG (150 kDa), thus at least suggesting that it could possibly be internalized via the same hydrodynamic drag mechanism. If so are other complement components also internalized?

To address this issue, a number of possible experiments could be done. Using immunofluorescence to show that Factor H can be detected at the surface of the cells (but not the knockout cells) would be ideal. I have to assume this was tried with no success. Have the authors tried a Proximity Ligation Assay? Perhaps using the FHR moAb and tagged FH (or FH domains) and FHR^{-/-} cells as the negative control would work.

We agree with the reviewer that it is important to show that FHR interacts with FH directly, and we have now done that and included the data in the manuscript in Fig. 5b and addressed this in the results section lines 221-231.

First, to answer the above questions, FH is a highly abundant serum protein, with different levels cited depending on the studies, but with levels as high as 5 µM in normal humans (De Paula *et al.*, 2017). It is also abundant in bovine serum (Mhatre and Aston, 1987). This is now stated in lines 110-112.

In our work, full length FH interacted with FHR over the whole range of concentrations used in the SPR experiments from 1 µM down to 0.03 µM, and the ~0.15 µM affinity of FHR for FH means that the receptor would be saturated or close to saturation depending on the serum and environmental conditions of the animal, as stated in lines 110-112.

We have now shown that 0.1 µM fluorescent bovine FH binds FHR^{+/+} and not FHR^{-/-} cells when added directly to live, cultured PCFs *in vitro*. We performed this experiment in the presence of 10 % heat inactivated foetal calf serum so there may have been some competition for binding from serum FH, as FH is pulled down by the FHR from fetal calf serum (Supplementary Fig. 4a,b). We did not wash out serum prior to ligand addition or before fixation, as stated in the methods lines 704-709.

We used PCFs as we have identified a complement C3 receptor expressed uniquely in both slender and stumpy BSFs which complicates interpretation of FH binding and uptake due to C3 bound FH. In BSFs, we would expect the FH to be rapidly internalised following hydrodynamic drag towards the flagellar pocket. It would be expected that other complement components could follow the same route if the protein they are bound to is freely diffusible on the trypanosome surface. The additional experiment provided above shows that both FH and the FHR-moAb can bind over the PCF cell surface and some internalised in PCFs.

Therefore, we have now shown that FHR functions to bind ligand in live cells, further supporting the conclusions within this manuscript.

References:

- de Paula, P. F. *et al.* Ontogeny of complement regulatory proteins - concentrations of factor h, factor I, c4b-binding protein, properdin and vitronectin in healthy children of different ages and in adults. *Scand. J. Immunol.* **58**, 572–577 (2003).
- Mhatre, A. & Aston, W. P. Isolation of bovine complement factor H. *Vet. Immunol. Immunopathol.* **14**, 357–375 (1987).

2) I will acknowledge that my ability to interpret these mathematical models that supposedly support the data is limited. I do not know why the authors did not use PAD1 as a marker of short stumpy, please clarify. Nevertheless, in order to suggest that these knockout lines are attenuated in mice and during transmission due to a susceptibility to complement, why not utilize any of the number of murine models that have disrupted complement pathways? The differences in parasitemia and stumpy life duration should disappear if the same experiments were performed using the C5^{-/-} mice from Jackson, for example (B10.D2-Hc0 H2d H2-T18c/oSnJ).

Please see our response to Reviewer 2, point 4 for details on further experiments we performed using mouse FH and *Cfh*^{-/-} mouse plasma (Supplementary Fig. 4 and Table 4). With respect to the mathematical model itself, we have now provided a comprehensive and detailed explanation of the model. We have included it in our re-submission as Supplementary Data 1, which also contains the links to the publicly available code for the model on GitHub. We stand by the model as providing a sound explanation for our observations. However, we do appreciate possible confusion and ambiguity surrounding its use and so we have changed the manuscript text in lines 251-285 to reflect this, to make it clear that the model was used to only support our observations within the context of our tsetse fly work.

The rationale for using this mathematical model was as follows. We observed a small difference in the parasitaemia for WT FHR^{+/+} and two FHR^{-/-} clones in mice, although all cell lines were able to survive. This observation is in keeping with our new experiments on weak binding of FHR to mouse FH (see Reviewer 2, point 4 and Supplementary Fig. 4 and Table 4). We applied a well-established mathematical modelling approach in the trypanosome field (MacGregor, Savill, Hall and Matthews, 2011; Dewar *et al.*, 2018) to provide an explanation for this observation, rather than setting out to measure slender and stumpy BSF levels from the outset. Our focus was on the effect of FHR on transmission to tsetse flies, and we wanted to see whether our observations in mice were consistent with the role of FHR as increasing transmissibility of infection.

We did not set the FHR^{-/-} stumpy lifespan to a lower value than FHR^{+/+}, but rather, lifespan was predicted from the model. We fit the model to our data and the outcome were a number of parameter estimates. The major observed difference in the model fits was the generally higher stumpy peaks in WT FHR^{+/+} mice than in FHR^{-/-} mice and this resulted from a predicted longer lifespan of stumpy forms in WT FHR^{+/+} mice compared to FHR^{-/-} mice. Simply put, the only significant difference in all parameter estimates between FHR^{+/+} and FHR^{-/-} mice was in stumpy duration.

As the reviewers point out, our manuscript contains a wide range of experiments. While we agree that PAD1 is a valid method for estimating stumpy levels, it was beyond the scope of this paper at this stage and would be an excellent future experiment for further studies on this receptor. We believe that at this point, and in particular because our main finding was on transmission to tsetse flies, that the mathematical model is sufficient to support our conclusions.

With respect to the use of complement C5^{-/-} mice or other complement ^{-/-} (e.g. C3^{-/-}) mice that are routinely available, C5 and C3 are also integral to the classical pathway. Therefore, these ^{-/-} mice are more susceptible to trypanosome infection as trypanosomes are very readily killed by antibody-mediated lysis by the classical pathway at high antibody titres (see Fig. 2a in Liu *et al.*, 2019). We would not be able to discern any FHR-specific effects. We considered using FH^{-/-} mice, but this was financially prohibitive (further explanation on FH^{-/-} mice to Reviewer 4, point 3). However, we did obtain a small quantity of *Cfh*^{+/+} and *Cfh*^{-/-} mouse plasma and observed a response only for *Cfh*^{+/+} and FHR (Supplementary Fig. 4).

References:

- Liu, G. *et al.* CR1g plays an essential role in intravascular clearance of bloodborne parasites by interacting with complement. *Proc. Natl. Acad. Sci. U. S. A.* **116**, 24214–24220 (2019).
- MacGregor, P., Savill, N. J., Hall, D. & Matthews, K. R. Transmission stages dominate trypanosome within-host dynamics during chronic infections. *Cell Host Microbe* **9**, 310–318 (2011).
- Dewar, C. E. *et al.* Mitochondrial DNA is critical for longevity and metabolism of transmission stage *Trypanosoma brucei*. *PLoS Pathog.* **14**, e1007195 (2018).

2) (continued) Lastly, the simplest possible experiment to do would be to isolate stumpy form parasites from both the WT and knockout lines and do *in vitro* complement lysis experiments with murine serum derived from WT mice as well as complement inactivated (ideally genetically) mice. At least this would be a financially more feasible option.

We considered this experiment in the past in great detail, as it is an obvious test. Preliminary experiments show that both cell lines are largely resistant to complement in a standard assay over 60 minutes. This would be expected as the predicted lifespan of stumpy BSFs reduced from 27 in FHR^{+/+} to 7 hours in FHR^{-/-} *in vivo*. The key experiment would be to directly determine the lifespan of stumpy BSFs. This has only been performed in culture, but this was done in the absence of complement (Dewar *et al.*, 2018). It is hard to envisage how this could be readily done in the animal model.

There are two factors that make measurements of stumpy half-life in response to complement technically difficult in this case. First, complement activation occurs rapidly and spontaneously *in vitro*, and even more quickly in the presence of activating agents and increasing temperature (Mollnes, Garred and Bergseth, 1988). As the BSF surface rapidly activates complement by the alternative and classical pathways (Devine, Falk and Balber, 1986; Engstler *et al.*, 2007), complement will also immediately begin to decay. It is hard to envisage how to maintain *in vivo* levels of complement in an *in vitro* experiment (half-life = minutes) over the time scale necessary to accurately measure a difference between FHR^{+/+} and FHR^{-/-} (half-life = hours). Second, the transition from slender to stumpy BSFs occurs over several cell cycles (Tyler, Matthews and Gull, 2001) and this poses great complexities in determining half-life *in vivo*.

As we explained above, as our focus here was on the role of FHR in transmission, and we have now added additional data to show a direct interaction between FHR and FH *in vitro* (Fig. 5b), we feel that further complement analyses are of value, both *in vitro* and *in vivo*, they are beyond the scope of this work. Again, we have minimised conclusions regarding stumpy BSFs to match this.

References:

- Devine, D. V., Falk, R. J. & Balber, A. E. Restriction of the alternative pathway of human complement by intact *Trypanosoma brucei* subsp. *gambiense*. *Infect. Immun.* **52**, 223–229 (1986).

- Dewar, C. E. *et al.* Mitochondrial DNA is critical for longevity and metabolism of transmission stage *Trypanosoma brucei*. *PLoS Pathog.* **14**, e1007195 (2018).
- Engstler, M. *et al.* Hydrodynamic flow-mediated protein sorting on the cell surface of trypanosomes. *Cell* **131**, 505–515 (2007).
- Mollnes, T. E., Garred, P. & Bergseth, G. Effect of time, temperature and anticoagulants on in vitro complement activation: consequences for collection and preservation of samples to be examined for complement activation. *Clin. Exp. Immunol.* **73**, 484–488 (1988).
- Tyler, K. M., Matthews, K. R. & Gull, K. Anisomorphic cell division by African trypanosomes. *Protist* **152**, 367–378 (2001).

Minor comments

1) I am curious as to what other molecules from Bovine serum will bind to this receptor. There are at least a couple more unique bands (albeit with lower abundance) present in figure 1A. The authors have indicated that they performed mass spec on the bands. Can we see the raw data from that experiment in the supplementary materials?

We only cut out the band of interest which was identified as bovine FH, as described in the methods section. Therefore, we have no further data on other bands present in the pulldown sample. The raw data is in the Source Data file.

2) Page 9, lines 25 and 26. Fix sentence.

This has been fixed.

3) Page 10, line 15: “reduced but did NOT abolish” is what I believe is intended.

This has been fixed.

4) Conceptually, it is surprising that the receptor is not expressed by slender BSFs. I wonder if the authors can expand on this in the discussion section. They currently have one or two sentences on the topic, wherein they suggest that the hydrodynamic clearance of complement might be sufficient for slender forms to evade lysis.

“How do slender BSFs protect themselves against the alternative complement pathway with little or no FHR expression? The process of hydrodynamic flow coupled to endocytosis to rapidly clear surface-bound antibodies 25 24 may be sufficient in slender BSFs to remove deposited C3b or C3bBb before downstream assembly of MAC occurs. It is also probable that there are additional mechanisms to counteract complement.”

However, the 2007 Engstler *et al.*, paper also shows that stumpy clear IgG from their surface approximately twice as fast as slenders if I'm not mistaken. This would suggest that the slenders are actually less efficient at clearing surface bound immune complexes such as deposited C3b.

We also found it surprising that the receptor is not expressed in BSF form trypanosomes, or is expressed at low levels (e.g. the HpHbR is expressed at 200 copies/cell). There are two possible explanations for this: 1) the slender BSF is able to internalise and inactivate C3b at a rate sufficient to prevent MAC formation in most cells and/or 2) there is an additional mechanism of inactivation possibly an alternative FHR. There is no obvious orthologue in the genome and this is a future investigation. Please see our response to reviewer 2, points 1-3 and the discussion lines 298-336, which provides further details on these questions and covers the work done by Engstler *et al.*, 2007.

Response to reviewer 4

Macleod et al. report the identification of a *Trypanosoma brucei* receptor for factor H (fH), a downregulator of the complement system. The authors demonstrate interaction between this trypanosomal receptor, called FHR, and bovine fH by pulldown and crosslinking experiments using bovine serum, and direct interaction between the purified forms of these proteins by surface plasmon resonance. The latter shows that the interaction between FHR and fH is quite tight, with a KD of 150 nM. The authors also identify several FHR amino acids as candidates for fH-interaction, and show that some of these are essential for fH-binding; a mutant FHR protein containing substitutions at these amino acids provides a valuable negative control. The authors also identify fH domain 5 as sufficient for interaction with FHR, and determine the X-ray crystal structure of fH domain 5 bound to FHR. Structural modeling of the intact fH-FHR complex suggests that fH remains free to interact with and downregulate the C3 convertase.

However, whether this indeed occurs is not tested (Major Point #1). The authors move from these in vitro studies to several in vivo ones. For the latter, they construct several independent FHR^{-/-} *T. brucei* lines. The authors determine that FHR is expressed, as detected by antibodies, in the stumpy blood stream form but not in the slender blood stream form, which developmentally precedes the stumpy form. The greatest expression of FHR, however, occurs in the procyclic form, which occurs in the midgut of tsetse flies. It wanes in the mesocyclic form, which migrate from the midgut to the proventriculus of tsetse flies, and is not detectable in the epimastigote form, which migrate from the proventriculus to the salivary gland of tsetse flies. The authors also show that FHR is accessible on the surface of the procyclic form using a monoclonal antibody. These results suggest a role for FHR in transmission between infected mammalian hosts and tsetse flies, which the authors confirm through mouse infection experiments.

However, the authors do not confirm that FHR interacts with murine fH domain 5, as opposed to bovine fH domain 5 as in the in vitro experiments (Major Point #2). Mild differences are seen in parasitemia caused by wild-type vs. FHR^{-/-} *T. brucei*, but based on an apparently well-established mathematical model, the authors draw the conclusion that loss of FHR results in a ~2-fold decrease in the stumpy blood stream form due to a decrease in lifespan of this form from ~27 h to ~7 h. It is remarkable that a mild effect in parasitemia can be modeled into a major change in lifespan (Other Issues #1). Lastly, the authors find that loss of FHR leads to a ~2.5-fold decrease in the number of *T. brucei* transmitted from infected mice into the midgut of tsetse flies. The work is solid overall, but indirect in places where direct evidence should be presented. Several major points could be addressed by the authors to strengthen the work.

We thank reviewer 4 for their insightful and detailed reading of our work and have replied to their comments specifically below.

Major Points

1) It is surprising that the authors do not directly examine the functional effect of recruitment of fH by FHR. For example, is the amount of C3b lower (or C3dg or C3d greater) on the surface of wild-type vs. FHR^{-/-} *T. brucei*?

We agree that it is important to show the functionality of the trypanosome FHR, and we have now shown that the receptor is able to directly recruit factor H in live cells in culture (Fig. 5b). Therefore, we have provided evidence that it is indeed a functional receptor. It has been shown by Devine, Falk and Balber (1986) that C3b and C3bBb are able to bind and assemble on the BSF surface, but that they are rapidly restricted and do not progress to the formation of the MAC. We have preliminary data that there is also a C3 binding protein in

slender and stumpy BSFs (see Reviewer 2, points 1-3, and Reviewer 3, point 1), which therefore makes measurements of C3dg and C3d levels as a direct response to FHR only technically difficult.

References:

Devine, D. V., Falk, R. J. & Balber, A. E. Restriction of the alternative pathway of human complement by intact *Trypanosoma brucei* subsp. *gambiense*. *Infect. Immun.* **52**, 223–229 (1986).

2. The *in vitro* experiments are carried out with bovine fH, while the *in vivo* experiments are carried out in mice. Therefore, it is crucial for the authors to verify that FHR interacts with murine fH domain 5. (Along these lines, it would also be useful to know whether FHR interacts with human fH domain 5.)

We have done further experiments to investigate FHR binding to FH from a range of mammals, including mouse and human, and all this data is within Supplementary Fig. 4 and Supplementary Table 4. Please see our comment to Reviewer 2, point 4, as well as the results section lines 175-197 and the discussion section lines 345-354.

3. A clear test of the role of fH and FHR in transmission would be to examine fH-knockout mice. Is there a reason why this was not done?

At the time, FH^{-/-} were not available as mice, but would have to be regenerated from frozen embryos. Although this is a good experiment, we chose here to not perform this for financial reasons, both regeneration of the mouse line and introduction into a tsetse fly facility. Further, the effect on transmission was confirmed in two separate clones and it is hard to envisage that this was a result of anything other than the deletion of the FHR gene. As a financially feasible option, we have tested *Cfh*^{+/+} and *Cfh*^{-/-} mouse plasma for binding to FHR and saw a response specific to *Cfh*^{+/+} (Supplementary Fig. 4f).

Other Issues

1. As general readers will likely have to take the mathematical modeling by faith, the authors should provide some explanation of the model and evidence of its predictive power. They should also show how the other parameters in the model behave between WT and FHR^{-/-} *T. brucei*.

We agree that this would strengthen our conclusions and we have done this. Please see our response to Reviewer 3, point 2 for a detailed answer to this question. We have now provided the full code in Supplementary Data 1. We have also lessened our claims about the model and stumpy BSFs, which can be found in results section lines 251-285.

2. A fuller explanation of the choice of Tb927.5.4020 would be helpful. Why was Tb927.5.4020 chosen from all the proteins listed in Table S1? Additionally, why were only GPI-anchored proteins considered? Why weren't transmembrane proteins considered?

This protein gave the most clear-cut pulldown results and so was the first target for which we proceeded to ligand identification and characterisation. Transmembrane proteins were considered in a wider screen and will be published separately.

3. The CD analysis of FHR mutant proteins establishes that structure is conserved (at whatever temperature the experiment was done, apparently not listed), but does not establish whether the mutant proteins are as stable as wild-type FHR. The authors should do a thermal melt or chaotropic denaturation of mutant and wild-type FHR, as monitored by CD.

We performed the CD analysis at 20 °C and this has been added to the methods. We have also performed a thermal melt of the WT and two mutant proteins as monitored by CD, and they have equal stability. These results have been added to Supplementary Fig. 3 and the raw data in the Source Data file. From these results, it is clear that all proteins were stable over the range of temperatures used in the experiments and temperatures within the tsetse fly and bloodstream.

4. Why are only approximate affinities given for the surface plasmon resonance analysis? Presumably the data yield a specific value with associated errors. These should be stated, and the fits to the data should be shown.

We do have the precise affinity values and fits and have now included these throughout the entire manuscript for all SPR experiments (Supplementary Figs. 1, 3, 4 and Source Data). We have now added the precise affinity values in the main text that match the data that is presented in all figures.

5. The purity of the various FH domains in Figure 2 should be shown. This is especially important for FH 7-9 and 18-20, as these are negative in the assay.

SDS-PAGE analysis of bovine FH domains 4-5-6, 7-8-9, 18-19-20, and FHR and bovine FH domain 5 have been included in Supplementary Fig. 3a, as well as for mouse and human FH domains 4-5-6 in Supplementary Fig. 4c.

6. There are a surprisingly large number of waters for a 2.7 Å resolution structure. Might this explain the rather large difference between R-free and R-work? Scant information is given about how the structure was refined.

We thank the reviewer for their question about the water molecules and the divergence of R_{work} and R_{free} . We have checked both. We confirm that the water molecules all meet our criteria for confidence, with a clear density and placement within a reasonable hydrogen bonding distance of an appropriate protein group, or a water attached to a protein group. We are therefore confident that they are appropriately attributed. The large number of water molecules may relate to the high exposed solvent surface area of this protein. Indeed, our structure of the *T. congolense* haptoglobin-haemoglobin receptor, which adopts a very similar fold, had 265 water molecules for 233 protein residues (albeit at 1.6 Å resolution) and so this structure presented in this manuscript is not out of line. On the issue of the divergence of R_{free} and R_{work} , we do not have a clear explanation, other than to note that we see the same effect in different refinement programs, including Buster, phenix and refmac, and that this is not altered by the inclusion of NCS parameters or target restraints, to prevent over-fitting. However, we see no reason for concern and are extremely confident about the quality of the structure and the biological conclusions which we draw from it.

7. The concentrations of proteins used for GST pulldowns should be detailed.

We have included this in the manuscript in the methods section lines 499-504.

8. The text could use a good dose of editing. There are a number of ungrammatical constructions and some careless mistakes. Some examples follow, but there are a number of others.

a. "The mechanisms by which *T. brucei* counteracts the mammalian adaptive immune response are well characterised, antigenic variation at the population level and rapid clearance of surface-bound immunoglobulin at the individual cell level^{24–26}."

b. "This work started with an assumption that the interactions between trypanosomes and their hosts are likely to be more extensive than those previously characterised, a screen of the *T. brucei* genome was performed to identify putative receptors based on one or both of

two criteria.”

c. It has been shown that C3bBb assembly occurred on the surface of BSFs but did the process did not progress further there there was no formation of a MAC by recruitment of complement proteins C5b-930.”

We have now made these changes and have done our best to fix all mistakes in the manuscript.

Reviewers' Comments:

Reviewer #3:

None

Reviewer #4:

Remarks to the Author:

The authors have addressed my prior concerns adequately. The points below are minor ones that the authors should consider in a final version of the paper.

1. "This concentration would enable receptor saturation at the measured affinity." A concentration ten-fold higher than KD would yield about 90% saturation, so "near saturation" would be more accurate.
2. "Circular dichroism (CD) spectroscopy was used to confirm that the changes made in mutants 2 and 3 had not substantially altered the structure (Supplementary Fig. 2a-d)." This is only part of the story. Mutant 3 appears to be substantially destabilized. This of course decreases the ratio folded to unfolded protein at all temperatures for mutant 3 compared to wild-type, and makes the interpretation of the chemical cross-linking experiment less straightforward than the authors present. This complication should be acknowledged.
3. Supplementary Figure 3. The fH 4-5-6 sample appears to have a substantial ~12 kDa contaminant. Presumably this contaminant makes estimation of the concentration of fH 4-5-6 problematic and therefore the KD value suspect.
4. Supplementary Table 4 has a rather subjective and perplexing grouping of amino acids. For Lys (yellow) in the bovine sequence "SKY," how would Ile (green, mouse) perform "some of total interactions" if the interaction is hydrogen bonding? And for Asp in the bovine sequence "TDA," why aren't Glu in sheep and goat colored gray (can perform same interaction)?

REVIEWERS' COMMENTS:

Reviewer #4 (Remarks to the Author):

The authors have addressed my prior concerns adequately. The points below are minor ones that the authors should consider in a final version of the paper.

1. "This concentration would enable receptor saturation at the measured affinity." A concentration ten-fold higher than K_D would yield about 90% saturation, so "near saturation" would be more accurate.

We have modified this sentence in line with this point.

2. "Circular dichroism (CD) spectroscopy was used to confirm that the changes made in mutants 2 and 3 had not substantially altered the structure (Supplementary Fig. 2a-d)." This is only part of the story. Mutant 3 appears to be substantially destabilized. This of course decreases the ratio folded to unfolded protein at all temperatures for mutant 3 compared to wild-type, and makes the interpretation of the chemical cross-linking experiment less straightforward than the authors present. This complication should be acknowledged.

All crosslinking experiments were performed at room temperature. At this temperature, WT and mutant 3 FHR have the same ellipticity. We agree that mutant 3 FHR has a slightly earlier melting temperature, but this does not affect our experiment. We have added "at the temperatures at which the experiments were performed" at the end of this sentence to clarify this point in the manuscript.

3. Supplementary Figure 3. The fH 4-5-6 sample appears to have a substantial ~12 kDa contaminant. Presumably this contaminant makes estimation of the concentration of fH 4-5-6 problematic and therefore the K_D value suspect.

We thank the reviewer for pointing this out. We have now clarified in the legend for Supplementary Figure 3 the following. These samples were at the initial stage of protein purification. Recombinant domains 4-5-6, 7-8-9, and 18-19-20 in these lanes were purified by nickel chromatography from CHO cell culture supernatant and dialysed. As the recombinant proteins also contained an AviTag, we biotinylated them to confirm expression. These samples were the ones analysed here by SDS-PAGE. The uncropped SDS-PAGE gel can be found in Source Data. After this initial stage, proteins were gel filtered and used for SPR. Therefore, the contaminant was not of concern in the final stages.

4. Supplementary Table 4 has a rather subjective and perplexing grouping of amino acids. For Lys (yellow) in the bovine sequence "SKY," how would Ile (green, mouse) perform "some of total interactions" if the interaction is hydrogen bonding? And for Asp in the bovine sequence "TDA," why aren't Glu in sheep and goat colored gray (can perform same interaction)?

We agree with the reviewer that this table required clarification and we have now done so. Please see our new version of Supplementary Table 4 as it should answer these two questions. While we agree that this table is somewhat subjective, crystallisation of domain 5 from each mammal was outside the scope of this work. We thus made our best assumptions as to whether different amino acids could perform the same interaction as bovine domain 5. In line the reviewer's comments, it should be noted that we removed one "conserved" interaction in horse, rabbit, rat and mouse, making the "total conserved interactions" for each now 13, 9, 13, and 11 (as opposed to 14, 10, 14, 12 in the prior manuscript). We have changed this in the final revised manuscript, but this does not change the conclusions whatsoever.